15

# Understanding spatio-temporal patterns of the propagation characteristics across meteorological, hydrological, and agricultural droughts and their influencing factors

Yuanrui Liu<sup>1</sup>, Tingting Hu<sup>1</sup>, Jiawen Yang<sup>2</sup>, Lei Yu<sup>1\*</sup>

<sup>1</sup>School of Water Conservancy and Transportation, Zhengzhou University, Zhengzhou, Henan, China <sup>2</sup>School of Environmental and Municipal Engineering, North China University of Water Resources and Hydropower Correspondence to: Lei Yu (yulei2018@zzu.edu.cn)

**Abstract.** Understanding the propagation of diverse drought conditions is necessary for drought preparedness. This study conducted a comprehensive analysis of the propagation characteristics across meteorological, hydrological, and agricultural droughts from 1958 to 2024 over global land areas, based on an ensemble of ERA5, GLDAS, and TerraClimate datasets. Using standardized drought indices at different accumulation periods, three drought propagation characteristics, including response time (RT), propagation rate (PR), and lag time (LT), were examined based on time-lag correlation analysis and multi-threshold run theory. The climatic and geographical feature factors that influence drought propagation were quantitatively evaluated using a SHapley Additive exPlanations (SHAP)-based attribution method. The results demonstrate the propagation pathways of meteorological-hydrological-agricultural drought at the global-scale, with the average RT, PR, and LT from meteorological to hydrological drought at 5.0 months, 55.3%, and 1.23 months; from meteorological to agricultural drought at 8.7 months, 30.3%, and 2.60 months; and from hydrological to agricultural drought at 5.8 months, 35.0%, and 2.49 months, respectively. Notable temporal and spatial heterogeneities are observed in the drought propagation characteristics, which are closely influenced by with the regional climatic feature. Globally, temperature and potential evapotranspiration are the primary factors influencing the propagation of meteorological drought to hydrological drought, whereas precipitation plays a decisive role in the propagation from meteorological or hydrological drought to agricultural drought. The findings underscore the importance of taking climatic characteristics into account in the development and implementation of regional drought risk management.

## 1 Introduction

Drought is one of the most frequent natural disasters and is generally defined as a prolonged period of moisture deficits within the water cycle (Liu et al., 2020; AghaKouchak et al., 2023). Under global warming, the magnitude, frequency, and spatial extent of droughts have significantly increased in recent decades, primarily as a result of rising atmospheric evaporative demand (Chen et al., 2025; Gebrechorkos et al., 2025). The intensified droughts pose a significant threat to ecosystems and socio-economic sectors, such as agricultural production (Hendrawan et al., 2022), ecosystem productivity



(Cao et al., 2022; Gu et al., 2025), and water resources (Fowler et al., 2022; Liu et al., 2025). Moreover, multi-model ensemble of climate and hydrological projections reveals a consistent drying trend across many regions during the 21st century (Cook et al., 2020; Christian et al., 2023; Li et al., 2025). The increasing frequency and intensity of droughts are expected to exacerbate water scarcity and ecosystem degradation in the foreseeable future, thus posing significant threats to both the natural environment and human society. Therefore, characterizing the spatio-temporal dynamics of droughts is of crucial importance, as it has significant implications for adapting to and mitigating the impacts of drought-related hazards.

Drought is a complex and multifaceted natural phenomenon within water cycle (Wu et al., 2022). Although a drought event typically originates from inadequate precipitation and excessive evapotranspiration (referred to as meteorological drought), its impacts on human and natural systems are closely related to subsequent development, such as diminished runoff (hydrological drought), and reduced soil moisture (agricultural drought). The transition processes from one type of drought (i.e., meteorological drought) to another (i.e., hydrological drought) are commonly referred to as drought propagation (Apurv et al., 2017; Colombo et al., 2024). Understanding drought propagation characteristics is essential for elucidating drought occurrence and evolution mechanisms, which help facilitate the effective drought monitoring and early warning systems. Over the past decades, numerous studies have investigated the propagation characteristics of various types of drought at both regional (Aryal et al., 2024; Geng et al., 2024) and global scales (Zhu et al., 2021, Shi et al., 2022; Liu et al., 2023), using methods such as correlation analysis (Han et al., 2023), run theory (Xiong et al., 2025), hydrological models (Gevaert et al., 2018; Yang et al., 2024), copula functions (Wu et al., 2022; Yang et al., 2024), causality analysis (Shi et al., 2022), complex network theory (Konapala et al., 2022) and machine learning models (Muthuvel and Qin, 2025). For example, Shi et al. (2022) examined the response time from meteorological and hydrological droughts using the maximum correlation coefficient method, and analyzed the variations in response time across different climatic regions. Han et al. (2023) investigated the propagation pathways of various types of droughts across China and revealed the long-chain propagation mechanisms involving meteorological, hydrological, agricultural, and groundwater droughts. Aryal et al. (2024) evaluated the propagation time from meteorological drought to hydrological and agricultural drought across Australia, emphasizing the significant influence of climatic conditions and drought indices in assessing drought propagation dynamics. Among the aforementioned studies, correlation analysis and run theory are two of the most commonly used methodologies for quantifying drought propagation from a statistical perspective (Zhang et al., 2022). The time-lagged correlation analysis based on standardized drought indices (SDIs) with varying accumulation periods provides a direct and effective method for assessing the response time among different drought types. However, the correlation analysis can only reflect the average linear relationship between different drought types, but cannot capture the variations among individual drought events (Zhou et al., 2024; Brunner and Chartier-Rescan, 2024). In comparison, the run theory identifies discrete drought events based on the time series of drought indices, thereby providing a more physically meaningful interpretation of the time lag relationship among various drought conditions. Although extensive research has been conducted to examine the characteristics of drought propagation, the results of these studies are heavily dependent on the datasets, evaluation methods, drought indices,







and thresholds employed. Consistent and comparable drought propagation assessment is desired for improving our understanding of drought propagation, particularly at a global scale.

The propagation characteristics of different types of drought vary depending on climatic conditions and underlying surface conditions (Apurv et al., 2017; Sattar et al., 2019; Apurv and Cai, 2020; Liu et al., 2023). Previous studies shown that the drought propagation is highly related to the climatic and geographical factors (Gevaert et al., 2018; Liu et al., 2023). Limited research has been conducted on quantifying the impacts of these factors on drought propagation. Data-driven machine learning approaches are increasingly being employed in drought modeling due to their inherent advantages in capturing nonlinear patterns from complex and high-dimensional data (Sundararajan et al., 2021; Prodhan et al., 2022). Moreover, the SHapley Additive exPlanations (SHAP) provides a unified attribution framework for explaining the ML outputs, enabling the interpretation of the causal relationships between independent variables and dependent variables (Antwarg et al., 2021; Nohara et al., 2022). Recently, SHAP-based attribution models have been utilized to investigate drought dynamics across various temporal and spatial scales (Xue et al., 2024; Feng et al., 2025). To the best of our knowledge, there is currently a lack of studies that analyze the factors influencing drought propagation from the perspective of SHAP-based machine learning.

Reliable drought monitoring and an in-depth understanding of the underlying mechanisms depend on datasets that accurately describe variations in drought-related hydro-meteorological variables. For large-scale and global assessments, gauge observations and gauge-based gridded datasets are often constrained by limited spatial and temporal coverage, the occurrence of missing values, and challenges in data accessibility (Wang et al., 2020; Gebrechorkos et al., 2024). Numerous satellites, reanalysis, earth system models, and merged datasets have been developed, providing long-term and spatially continuous records of hydro-meteorological variables (Abatzoglou et al., 2018; Hersbach et al., 2020). Over the past decades, extensive efforts have been conducted to evaluate drought dynamics using different datasets at both regional and global scales. For example, Yuan et al. (2023) assessed the global patterns of flash drought, which is characterized by the rapid depletion of soil moisture, using ERA5 reanalysis and climate model datasets. Gebrechorkos et al. (2025) revealed the critical role of atmospheric evaporative demand in accelerating global drought severity, based on an ensemble of reanalysis, gridded observation, and hydrological model datasets. Wu et al. (2025) conducted an assessment of the dynamic predictability of agricultural drought across global land areas, utilizing the CRU gridded observation, ERA5 reanalysis, and GLDAS hydrological model datasets. However, inconsistent findings across studies can be attributed to the inherent uncertainties and errors within different datasets; few systematic comparisons have been conducted to quantify the discrepancies among the multiple datasets in representing drought propagation characteristics.

Although previous studies have evaluated drought propagation across various temporal and spatial scales, a comprehensive global assessment of the propagation characteristics of meteorological, hydrological, and agricultural droughts remains




lacking, and the underlying influencing factors are not yet fully understood. Therefore, the objectives of this study are as follows: (1) to assess the spatial and temporal patterns of response time, propagation rate, and lag time across meteorological, hydrological, and agricultural droughts; (2) to quantify the main climatic and underlying surface feature factors that influence drought propagation characteristics from the perspective of machine learning; (3) to evaluate the reliability and uncertainty associated with multi-dataset ensembles in drought propagation assessments.

## 2 Data and methodology

#### 2.1 Datasets

Datasets play a crucial role in characterizing drought conditions. For large-scale studies, in-situ stations for hydrometeorological variables are insufficient to cover all global terrestrial regions, and their temporal series are also limited in duration. To ensure the reliability of the drought propagation analysis, the hydro-meteorological variables (i.e., precipitation, potential evapotranspiration, runoff, and soil moisture) used to calculate drought indices in our study were derived from three different datasets, including ERA5, the Global Land Data Assimilation System (GLDAS), and TerraClimate. Multiple datasets not only avoid the bias relying on a single dataset, but also provide a more comprehensive understanding of drought propagation from various perspectives of water cycle modelling.

ERA5 is the fifth-generation global atmospheric reanalysis product developed by the European Centre for Medium-Range Weather Forecasts (ECMWF) (https://cds.climate.copernicus.eu/). It integrates extensive records of both in-situ and satellite observations through an ensemble-based data assimilation system (Hersbach et al., 2020). GLDAS is a multi-model ensemble comprising three land surface models—Noah, Catchment (CLSM), and the Variable Infiltration Capacity (VIC)—which integrate satellite and in-situ observations through advanced land surface modeling techniques (https://ldas.gsfc.nasa.gov/). TerraClimate is a high-spatial-resolution, merged hydro-meteorological dataset that covers global terrestrial surfaces for the period from 1958 to 2024 (https://doi.pangaea.de/10.1594/PANGAEA.909132). TerraClimate integrates multiple datasets, such as WorldClim, Climate Research Unit (CRU), and Japanese 55-year Reanalysis (JRA-55), to generate hydro-meteorological variables. Potential evapotranspiration in the TerraClimate is calculated using the Penman-Monteith method, while runoff and soil moisture are estimated through an empirical water balance model (Abatzoglou et al., 2018).

In addition, the Normalized Difference Vegetation Index (NDVI) was obtained directly from the Advanced Very High Resolution Radiometer (AVHRR) instruments operated by the National Oceanic and Atmospheric Administration (NOAA) (Pinzon and Tucker, 2014). The evaluation dataset was obtained from the ETOPO Global Relief Model developed by the National Centers for Environmental Information (https://www.ncei.noaa.gov/products/etopo-global-relief-model). The



aridity index dataset was derived from the Global Aridity Index and Potential Evapotranspiration Database—Version 3 (Zomer et al., 2022).

## 2.2 Calculation of drought indices

Effective drought monitoring relies on drought indices to detect various drought conditions. In this study, standardized drought indices (SDIs) derived from precipitation, runoff and soil moisture were employed to characterize meteorological, hydrological and agricultural droughts, as well as their propagations. The SDI time series were obtained by fitting the drought variables of interest to a suitable probability distribution and subsequently normalizing the probabilities to generate a standardized time series. Three SDIs, including Standardized Precipitation Index (SPI) (McKee et al., 1993), Standardized Runoff Index (SRI) (Shukla and Wood, 2008), and Standardized Soil Moisture Index (SSI) (Hao and AghaKouchak, 2013), were computed by fitting parametric probability distributions, specifically the Gamma, log-normal, and normal distributions, respectively. The maximum likelihood estimation (MLE) method was utilized to estimate the parameters of the probability distributions, with the initial values determined based on the L-moments estimation (Stagge et al., 2015). Compared with other drought indices, the SDI is not only simple and efficient to calculate, but also applicable to diverse climates due to its consistent calculation procedure. Meanwhile, SDI can be calculated using drought-related variables across multiple time scales, thus enhancing its effectiveness in analyzing drought propagation.

## 2.3 Response time analysis based on correlation coefficient

The response times among meteorological, hydrological, and agricultural droughts were analyzed using correlation analysis. Assuming that a high correlation coefficient indicates a strong relationship, the correlation analysis between drought indices with different accumulation periods can be conducted to determine the response time of different drought types (Zhang et al., 2022). For example, the response time from meteorological drought to agricultural drought is determined by the accumulation period of SPI that corresponds to the maximum correlation coefficient with the SSI at a 1-month accumulation period. A shorter accumulation period of SPI to 1-month SSI indicates a more rapid agricultural drought response to meteorological drought conditions. The correlation coefficient is calculated using Pearson's correlation coefficient formulated as follows (Pearson, 1896):

$$r_{P} = \frac{\sum_{i=1}^{n} (x_{i} - \overline{x})(y_{i} - \overline{y})}{\sqrt{\sum_{i=1}^{n} (x_{i} - \overline{x})^{2}} \sqrt{\sum_{i=1}^{n} (y_{i} - \overline{y})^{2}}}$$
(1)

where  $r_P$  represents the Pearson's correlation coefficient between SPI-n (n is the accumulation period, n = 1, 2, ..., n) and SSI-1;  $\overline{x}$  and  $\overline{y}$  represent the average value of SPI and SSI, respectively;  $x_i$  and  $y_i$  represents the SPI and SSI values in the time i, respectively. The Pearson's correlation coefficient is ranged from -1 to 1, with positive correlation with  $r_P > 0$ , and negative correlation with  $r_P 



 $(R_{MH})$ , from meteorological drought to agricultural drought  $(R_{MA})$ , and from hydrological drought to agricultural drought  $(R_{HA})$  by analyzing the correlations between SPI and SRI, between SPI and SSI, and between SRI and SSI, respectively.

## 160 2.4 Lag time analysis based on run theory

Run theory is a commonly used method for analyzing drought characteristics, which defines the initiation and termination of a drought event based on the drought index. In this study, the drought events were identified using the run theory of multiple thresholds (Fleig et al., 2006; Ma et al., 2021). Potential drought events were initially identified using an intermediate threshold ( $X_0 = 0$ ). Subsequently, the adjacent drought events with an interval of one month and whose drought index values were below a high threshold ( $X_1 = 1$ ) within that month were combined. Finally, the potential drought events with on month length and whose drought index value is greater than a low threshold ( $X_2 = -1$ ) were ruled out.

After using run theory to identify the initiation and termination of drought events, the propagation rate and lag time between the two types of droughts can be evaluated. Taking meteorological and agricultural droughts as an example, the propagation from meteorological drought to agricultural drought is defined as the occurrence of an agricultural drought event during the period in which a meteorological drought occurs. Thus, the propagation rate  $(P_{MA})$  and lag time  $(LT_{MA})$  can be mathematically expressed as follows (Sattar et al., 2019):

$$P_{MA} = \frac{n}{m} \times 100\% \tag{2}$$

$$LT_{MA} = \frac{\sum_{i=1}^{n} (T_{M,i} - T_{A,i})}{n}$$
(3)

where *n* is number of meteorological drought events that propagate to agricultural drought events; *m* is the total number of meteorological drought events during the study period; T<sub>M,i</sub> is the starting time of meteorological drought event *i*, and T<sub>A,i</sub> is the starting time of agricultural drought event *i*. Consistent with the analysis of drought response time, we analyzed the propagation rate and lag time between meteorological and hydrological droughts (P<sub>MH</sub> and LT<sub>MH</sub>), between meteorological and agricultural droughts (P<sub>MA</sub> and LT<sub>MA</sub>), and between hydrological and agricultural droughts (P<sub>HA</sub> and LT<sub>HA</sub>) by utilizing the SPI, SRI, and SSI at a one-month accumulation scale.

# 2.5 SHAP-based attribution analysis

The characteristics of drought propagation are closely associated with regional climatic and geographical features. In this study, we quantified the feature factors that influence drought propagation characteristics across global land areas using the SHapley Additive exPlanations (SHAP) method. SHAP is an effective method for interpreting the outputs of machine



learning models, which are often treated as black-box systems, based on principles derived from cooperative game theory (Nohara et al., 2022). The explanation model in the SHAP method can be represented as follows (Antwarg et al., 2021):

$$G(x) = \varphi_0 + \sum_{i=1}^{i-1} \varphi_i \tag{4}$$

where G(x) is the simulated drought propagation characteristics;  $\varphi_0$  is the average predicted drought propagation characteristics; and  $\varphi_i$  represents the SHAP value for feature factor i. For a machine learning model f and input instance x, the SHAP value  $\varphi_i$  for feature factor i can be formula as:

$$\varphi_{i}(f,x) = \sum_{S \subset N \setminus \{i\}} \frac{|S|!(|N| - |S| - 1)!}{|N|!} (f(S \cup \{i\}) - f(S))$$
(5)

where N is the set of all feature factors, and S is a subset of feature factor excluding feature factor i. The SHAP value  $\varphi_i$  can quantify the magnitude and direction of feature factors influencing on the model predictions. In our analysis, the drought propagation characteristics (i.e., response time, propagation rate and lag time) are simulated using the Extreme Gradient Boosting (XGBoost) model based on eight climatic and physiographical feature factors, including precipitation, temperature, potential evapotranspiration, runoff, soil moisture, aridity index, evaluation, and vegetation condition. The XGBoost model is an efficient and robust gradient-boosted decision tree algorithm that is widely applied in classification and regression tasks within the field of water resources engineering (Chen and Guestrin, 2016; Niazkar et al., 2024).

## 3 Results




## 3.1 Response time and correlation of drought indices

Figure 1. Spatial patterns of average  $RT_{MH}$ ,  $RT_{MA}$  and  $RT_{HA}$  across global land areas, and the corresponding maximum Pearson correlation coefficients. The blank grids indicate that the correlation between different drought indices is not statistically significant.

Figure 1 illustrates the spatial patterns of response times between meteorological, hydrological and agricultural droughts based on the ensemble of three datasets. The average RT<sub>MH</sub>, RT<sub>MA</sub> and RT<sub>HA</sub> across global land areas are 5.0 months, 8.7 months, and 5.8 months, respectively, with corresponding interquartile ranges (IQRs) being [2.7, 6.7] months, [5.0, 11.3] months, and [2.3, 7.3] months. The results indicate that hydrological drought responds more rapidly to meteorological drought compared to agricultural drought. Notable spatial heterogeneity is observed in the response times of various types of drought, with shorter propagation time in the regions of southeastern Asia, central Africa, South America, and North America. The maximum correlation coefficients reflect the reliability of the response time evaluation derived from

correlation analysis. Figures 1B, 1D and 1F display the spatial patterns of the maximum Pearson correlation coefficients, with the IQRs for SPI-SRI, SPI-SSI, and SRI-SSI being [0.43, 0.80], [0.51, 0.68], and [0.52, 0.70], respectively. The correlation coefficient between SPI and SRI in the mid- to low-latitude regions (30°S~30°N) is relatively high, suggesting a strong reliability in the response time from meteorological drought to hydrological drought in these areas. In comparison, aside from the high-latitude regions of the Northern Hemisphere, the correlation coefficients between SPI and SSI, as well as SRI and SSI, exhibit relatively strong correlations (> 0.5) across most regions.

Figure S1 illustrates the similar results based on different datasets, and the corresponding maximum Pearson correlation coefficients are presented in Figure S2. There are notable differences in response time and maximum correlation across different datasets, particularly in mid- to high-latitude regions. For the response time, consistent low RT<sub>MH</sub> values are observed in the TerraClimate dataset across global land areas, while high RT<sub>MH</sub> values are predominantly found in the high latitudes of the Northern Hemisphere. Among the three datasets, the relatively higher RT<sub>MA</sub> values are observed in the GLDAS dataset, while relatively higher RT<sub>MA</sub> values are found in the TerraClimate dataset. For the maximum correlation of RT<sub>MH</sub>, RT<sub>MA</sub>, and RT<sub>HA</sub>, the GLDAS dataset generally exhibits higher correlation coefficients, while the ERA5 and TerraClimate datasets display significant spatial heterogeneity. Particularly, the relatively low correlation coefficients (

Figure 2. Box plots of  $RT_{MH}$ ,  $RT_{MA}$  and  $RT_{HA}$  in each calendar month across global land areas, and the corresponding maximum Pearson correlation coefficients.

Figure 2 shows box plots illustrating the response times among meteorological, hydrological, and agricultural droughts in each calendar month, and the corresponding maximum Pearson correlation coefficients. The response time exhibits distinct fluctuations across various months, revealing a seasonal pattern of drought propagation. Shorter propagation times and higher correlation coefficients are observed during the period from June to September, which corresponds to the summer months in the Northern Hemisphere. The shortest response times for RT<sub>MH</sub>, RT<sub>MA</sub>, and RT<sub>HA</sub> are consistently recorded in August, with average values of 3.5 months, 5.6 months, and 3.6 months, respectively. In contrast, longer response times are observed during the colder seasons, with the longest response times for RT<sub>MH</sub>, RT<sub>MA</sub>, and RT<sub>HA</sub> occurring in February (5.6 months), April (7.7 months), and March (5.4 months), respectively. This result indicates the influence of seasonal temperature variations on the drought propagation patterns. For example, during the cold season, precipitation is typically retained in the form of snowpack, thereby delaying the propagation of meteorological drought to hydrological drought.


Figure 3. Spatial patterns of temporal trends in  $RT_{MH}$ ,  $RT_{MA}$  and  $RT_{HA}$  across global land areas. The blank grids indicate that the correlation between different drought indices is not statistically significant. The black dots indicate the statistical significance of the time series trend, where the p-values from the M-K test are less than 0.05.

Figure 3 presents the spatial patterns of the time series trends for RT<sub>MH</sub>, RT<sub>MA</sub>, and RT<sub>HA</sub>, calculated using 30-year moving windows based on the Sen's slope estimator and the M-K test. The temporal trends in response time across different drought types demonstrate notable spatial heterogeneity. The grids exhibiting a monotonic time series trend account for 53.4%, 35.1%, and 55.9% of the total grid for RT<sub>MH</sub>, RT<sub>MA</sub>, and RT<sub>HA</sub>, respectively. In contrast, the percentages of statistically significant increases range from 23.0% to 26.9%, whereas the percentages of statistically significant decreases fall within the range of 18.3% to 31.8%. For the RT<sub>MH</sub>, regions exhibiting increasing trends are primarily located in Europe, northwestern Asia, central Africa, and North America, while decreasing trends are sporadically observed across Asia. The decreasing trends of RT<sub>MA</sub> are primarily observed in the mid- to high-latitudes of the Northern Hemisphere, while increasing trends are predominantly evident in Central Asia and North America. In contrast, aside from the obvious increase observed in Central


Asia, no clear spatial distribution pattern can be identified in the time series trend of RT<sub>HA</sub>. This result indicates that the response time of various drought conditions can vary across different regions, exhibiting significant interannual variability.

## 3.2. Propagation rate and lag time of drought events

Figure 4. Spatial patterns of propagation rate ( $PR_{MH}$ ,  $PR_{MA}$  and  $PR_{HA}$ ) and lag time ( $LT_{MH}$ ,  $LT_{MA}$  and  $LT_{HA}$ ) across global land areas.

Figure 4 illustrates the spatial patterns of propagation rate and lag time across global land areas. The propagation rate represents the linkages between two kinds of drought events. A high propagation rate indicates that subsequent drought events are more sensitive to prior drought events. Among three pairs of drought propagation, the propagation rate from meteorological to hydrological droughts is highest, with a global average  $PR_{MH}$  value of 55.3% and an IQR of [46.4, 63.2]%. Spatially, a higher  $PR_{MH}$  (larger than 60%) tends to occur in mid- and low-latitude regions as well as in low-altitude areas, and is often associated with a shorter  $LT_{MH}$  (less than 1 month). The average  $LT_{MH}$  across global land areas is 1.23 months, with IQR of [0.68, 1.68] months. Consistent with the correlation analysis between SPI and SRI, the propagation from

meteorological drought to hydrological drought is significantly influenced by regional temperature conditions. Tropical and subtropical regions exhibit a higher propagation rate and a shorter lag time from meteorological drought to hydrological drought. The average PR<sub>MA</sub> and PR<sub>HA</sub> over global land areas are 30.3% and 35.0%, respectively, with the IQRs being [19.3, 41.5]% and [23.0, 47.6]%; and average LT<sub>MA</sub> and LT<sub>HA</sub> are 2.60 and 2.49 months, with the IQRs being [1.71, 2.92] and [1.68, 2.51] months. The higher PR<sub>MA</sub> and PR<sub>HA</sub> (larger than 40%) along with the shorter LT<sub>MA</sub> and LT<sub>HA</sub> (less than 2 month) trend to occur in humid regions, such as eastern North America, South America, central Africa, and southeastern Asia.




275

Figures S3 and S4 show spatial patterns of propagation rate and lag time in the ERA5, GLDAS, and TerraClimate datasets. Consistent with the results of the response time analysis, the highest  $PR_{MH}$  and  $LT_{MH}$  values are found in the TerraClimate datasets, with  $PR_{MH}$  values in the low and middle latitudes approaching 90% and  $LT_{MH}$  values approaching less than 1 month, respectively. In the ERA5 dataset, the relatively low  $PR_{MH}$  and high  $LT_{MH}$  values are found in the high latitudes of the Northern Hemisphere; while in the GLDAS dataset, the relatively low  $PR_{MH}$  and high  $LT_{MH}$  values are found in the hyperarid regions, such as Sahara and the Arabian Peninsula. The highest  $PR_{MH}$  value is detected in the ERA5 dataset, with an average value of 33.6% across global land areas, followed by GLDAS (30.2%) and TerraClimate (26.9%). The average  $PR_{MA}$  values are 2.05 months, 2.70 months, and 3.03 months for the ERA5, GLDAS, and TerraClimate datasets, respectively. This suggests that the ERA5 reanalysis reveal a significant global sensitivity of agricultural drought to meteorological drought. Similarly, agricultural drought exhibits greater sensitivity to hydrological drought, as evidenced by the simulations in the GLDAS dataset. The average  $PR_{HA}$  values are 36.4%, 43.5%, and 24.9% for ERA5, GLDAS, and TerraClimate datasets, respectively, while the corresponding average  $LT_{HA}$  values are 3.46 months, 1.87 months, and 2.08 months.


Figure 5. Spatial patterns of temporal trends in the  $PR_{MH}$ ,  $PR_{MA}$  and  $PR_{HA}$  across global land areas.

Figures 5 and 6 illustrate the spatial distributions of temporal trends in propagation rate and lag time across global land areas, based on a 30-year moving window analysis from year 1958 to 2024. The PR<sub>MH</sub> shows a significant decline across 49.0% of global land areas, while the corresponding LT MH exhibits a notable decrease in 47.8%. This indicates a reduced propagation from meteorological drought to hydrological drought in nearly half of the global land area. In contrast, the  $P_{MH}$  (LT  $_{MH}$ ) shows a significant increase across 28.1% (29.6%) of global areas, particularly in the North America and South America. Although more than 42.5% (50.9%) of the land area exhibits a significant decreasing trend in PR<sub>MA</sub> (LT<sub>MA</sub>), notable increasing trends are observed in the western North America, central South America, and central Africa. This result indicates that the sensitivity of agricultural drought to meteorological drought exhibits significant spatial heterogeneity. For the PR<sub>HA</sub>, the land areas experiencing increasing trends (significant percentages of 46.7%) are more extensive than those showing decreasing trends (significant percentages of 31.7%). Notable increase trends are observed in the western North America,

central South America, central Africa, Europe, and northern Asia. Similar spatial patterns are observed in the  $LT_{HA}$ , with significant increases and decreases accounting for 37.0% and 38.3%, respectively.

Figure 6. Spatial patterns of temporal trends in the  $LT_{MH}$ ,  $LT_{MA}$  and  $LT_{HA}$  across global land areas.


# 3.3. SHAP-based attribution of drought propagation characteristics

Figure 7. Scatter plots of SHAP values for feature factors influencing response time, propagation rate, and lag time across global land areas. (P-precipitation, T-temperature, PET-potential evapotranspiration, R-runoff, SM-soil moisture, AI-aridity index, DEM-evaluation, NDVI-vegetation condition).

Spatial analysis of drought propagation indicates that regional climatic conditions and physiographical characteristics significantly influence the patterns and dynamics of drought propagation. In this study, we conducted an in-depth analysis of the influence of climate, topography and vegetation conditions on the processes of drought propagation, using a SHAP-based attribution method. Figure 7 presents the SHAP values of eight feature factors that influence drought propagation characteristics across global land areas, including precipitation, temperature, potential evapotranspiration, runoff, soil moisture, aridity index, evaluation, and vegetation condition. The SHAP value reflects both the direction and magnitude of the influence of feature factors on drought propagation characteristics. A positive SHAP value indicates that the feature has contributed to an increase in response time, propagation rate, and lag time. For example, taking the influence of temperature


on the R<sub>MH</sub> as an example, the SHAP value indicates that high temperatures have shortened the response time of meteorological drought to hydrological drought (Figure 7A). The scatter plots of SHAP values indicate that precipitation, temperature, potential evapotranspiration, runoff, aridity index, and NDVI are all positively correlated with the three drought propagation characteristics. Particularly, the evaluation is negatively correlated with drought propagation characteristics, whereas soil moisture exhibits an inconsistent correlation with these characteristics.

Figure 8. Ranking of feature factors influencing drought propagation characteristics based on the absolute SHAP value.

Figure 8 presents the ranking of feature factors that influence drought propagation characteristics, based on the absolute SHAP values, which is consistent with the scatter plots of SHAP values. The greater the absolute value of SHAP, the more significant its influence on the characteristics of drought propagation. Temperature and potential evapotranspiration are the most significant factors that influence the propagation of meteorological drought into hydrological drought. The average


absolute SHAP values of temperature (potential evapotranspiration) for RT<sub>MH</sub>, PR<sub>MH</sub>, and LT<sub>MH</sub> are 0.65 (0.37), 3.34 (2.14), and 0.13 (0.24), respectively, ranking first (fourth), first (second), and second (first) among all influencing factors. This result indicates that in warm regions characterized by higher average temperatures and potential evapotranspiration, hydrological drought is more sensitive to meteorological drought, thus having a faster response time and a shorter lag time. In comparison, precipitation serves as the main influencing factor in the propagation from meteorological and hydrological droughts to hydrological drought. Among all feature factors considered for RT<sub>MA</sub>, RT<sub>HA</sub>, PR<sub>MA</sub>, PR<sub>HA</sub>, LT<sub>MA</sub>, and LT<sub>HA</sub>, precipitation exhibits the highest average absolute SHAP value. The differences in SHAP values among other feature factors are not obvious and vary depending on the distinct characteristics of drought propagation. In humid regions, agricultural drought trends to be more sensitive to meteorological and hydrological drought conditions.

Figure 9. Box plots of drought propagation characteristics across global land areas classified by the percentiles of key feature factors.







Figure 9 presents the boxplots of response time, propagation rate, and lag time across global land areas, grouped based on the percentiles of key feature factors identified through SHAP attribution analysis. The propagation from meteorological drought to hydrological drought is mainly influenced by regional temperature and PET. In the 20th to 70th percentiles of temperature and PET, both R<sub>MH</sub> and LT<sub>MH</sub> decrease as temperature and PET increase, while P<sub>MH</sub> increases as temperature and PET increase. This result indicates that as temperature and PET increase, hydrological drought becomes more sensitive to meteorological drought, and this relationship remains robust within the intermediate ranges of temperature and PET. For example, the average RT<sub>MH</sub> is 6.61 months in the 20th to 30th percentiles, decreasing to 3.43 months in the 60th to 70th percentiles; meanwhile, the average RT<sub>MH</sub> is 6.88 months in the 0th to 10th percentiles and 4.60 months in the 90th to 100th percentiles. In comparison, the propagations from meteorological and hydrological droughts to agricultural drought are mainly derived by the regional precipitation patterns. The average RT<sub>MA</sub>, RT<sub>HA</sub>, LT<sub>MA</sub>, and LT<sub>HA</sub> consistently decrease with increasing percentiles of precipitation, whereas the average PR<sub>MA</sub>, PR<sub>HA</sub> increase with increasing percentiles of precipitation. This indicates that agricultural drought driven by soil moisture deficits is more sensitive to meteorological and hydrological droughts in humid regions, showing a strong linear correlation with regional precipitation patterns.

#### 4. Discussion

## 4.1. Comparison of different propagation characteristics

Drought propagation is a critical characteristic of multiple drought conditions, which provides valuable information for monitoring and predicting drought dynamics. In this study, three drought propagation characteristics (i.e., response time, propagation rate, and lag time) across meteorological, hydrological, and agricultural droughts were comprehensively evaluated over global land areas. These characteristics reflect distinct aspects of the drought propagation signal within the hydrological cycle, as determined by various methodological approaches (Zhang et al., 2022). The response time is calculated by identifying the maximum Pearson correlation coefficient between two types of drought indices over various accumulation periods. This method facilitates a consistent comparison of drought propagation across diverse climatic regions and minimizes the subjectivity inherent in the evaluation of drought propagation. The response time also functions as an indicator of the feasibility of using one type of drought index as a proxy for another. For example, due to the lack of comprehensive observational data, the SPI with varying accumulation periods can reflect hydrological, agricultural and groundwater drought conditions (Kumar et al., 2016). In comparison, the propagation rate and lag time are calculated based on the drought events as defined by the multi-threshold run theory, which offers a more physically interpretable approach than correlation analysis. The event-based analysis incorporating threshold exceedances also captures nonlinear relationships among various drought conditions. The correlation analysis and run theory methods each possess distinct advantages and limitations. Our findings provide a systematic comparison of the consistency and discrepancies between these two approaches in characterizing the processes of drought propagation.







Consistently, the average response time ( $RT_{MH} = 5.0$  [2.7, 6.7] months,  $RT_{MA} = 8.7$  [5.0, 11.3] months,  $RT_{HA} = 5.8$  [2.3, 7.3] months), propagation rate ( $PR_{MH} = 55.3$  [46.4, 63.2]%,  $PR_{MA} = 30.3$ % [19.3, 41.5]%,  $PR_{HA} = 35.0$  [23.0, 47.6]%) and lag time ( $LT_{MH} = 1.23$  [0.68, 1.68] months,  $LT_{MA} = 2.60$  [1.71, 2.92] months,  $LT_{HA} = 2.49$  [1.68, 2.51] months) over global land areas highlight that hydrological drought responds more rapidly to meteorological drought compared to agricultural drought. Our findings demonstrate the propagation pathway of meteorological-hydrological-agricultural droughts at the global scale, which is consistent with previous findings on drought propagation at the regional scale (Han et al., 2023). This is consistent with the conceptual framework of drought propagation, where precipitation deficits (meteorological drought) first influence surface soil moisture and runoff (hydrological drought), and subsequently affect deeper soil moisture (agricultural drought). A similar global assessment of multiple types of droughts was also conducted by Liu et al. (2023), whose results indicate that the globally average  $RT_{MH}$  and  $RT_{MA}$  are 3.5 months and 5.7 months, respectively. This discrepancy arises from the inconsistent datasets utilized in characterizing response time of drought propagation. In our study, we rely on an ensemble of three advanced datasets recently developed, thereby providing more reliable estimates of response time across various drought conditions.

## 4.2. Main factors influencing drought propagation

Across the global land areas, the characteristics of drought propagation exhibit notable temporal and spatial heterogeneity. Regions such as South America, eastern North America, central Africa, and southeastern Asia demonstrate shorter response times, higher propagation rates, and longer lag times across meteorological, hydrological, and agricultural droughts (Figures 1 and 4). From year 1958 to 2024, 41.3% to 58.7% of global land areas exhibited significant temporal trends in response time, whereas 75.3% to 78.4% areas showed significant temporal trends in propagation rate and lag time (Figures 3, 5 and 6). In this study, we used XGBoost models with a SHAP-based attribution method to quantify the impacts of climatic, topographic, and vegetation-related feature factors on drought propagation. Our findings demonstrate that temperature and PET are the key factors influencing the propagation from meteorological to hydrological drought, while precipitation predominantly determines the propagation from meteorological/hydrological to agricultural drought. This is consistent with previous studies highlighting the significant role of climate characteristics in drought propagation (Apurv et al., 2017). In the tropical and subtropical regions (with temperatures and PET ranging from the 20th to the 70th percentiles), the increases in temperature and PET are expected to reduce both RT<sub>MH</sub> and LT<sub>MH</sub>, while simultaneously increasing PR<sub>MH</sub> (Figure 9). This is primarily attributed to the mechanism by which temperature influences the water cycle, leading to a lagged response of runoff to changes in precipitation. In cold regions and during cold reasons, precipitation is stored in the form of snow and glaciers, which subsequently melt and contribute to runoff during the following warm season. Therefore, the sensitivity of hydrological drought to meteorological drought is significantly influenced by temperature variations. In comparison, the sensitivity of agricultural drought to meteorological and hydrological droughts is closely associated with the regional average precipitation. This is because deep soil layers and aquifers in humid regions generally exhibit high saturation levels, where fluctuations in soil moisture demonstrate a more significant response to variations in precipitation and runoff.

## 4.3. Uncertainties in drought propagation evaluation

Due to the inherent variability of drought-related variables, significant uncertainties exist within hydrometeorological datasets (Bador et al., 2020). Our findings depend on an ensemble of three datasets (i.e., ERA5, GLDAS, and TerraClimate), which helps avoid biased and incomplete evaluations of drought propagation that could result from relying on a single dataset. We conducted a comparative analysis of drought propagation characteristics derived from multiple datasets, systematically evaluating their consistency and discrepancies (Figures 11-13). The results underscore the impact of input 425 data uncertainties on the assessment of drought propagation, with notable discrepancies predominantly observed in the hyper-arid, high-latitude, and high-evaluation regions. This is primarily attributed to the scarcity of in-situ stations capable of providing continuous spatial and temporal observations in these regions. The data assimilation systems and land surface models employed across different datasets to fill missing observations inevitably introduce uncertainties in both model parameters and structural configurations. Generally, this study provides a comprehensive assessment of multiple drought propagation characteristics across global land areas, which has significant implications for the development and 430 improvement of drought monitoring and early warning systems. In tropical and sub-tropical regions, real-time monitoring of meteorological drought can improve the forecasting of hydrological drought; whereas in humid regions, drought indices based on precipitation and runoff could provide more accurate predictions of agricultural drought risks. Future research could focus on improving the accuracy of predicting future drought changes by incorporating the mechanisms of drought 435 propagation into predictive models.

## 5. Conclusions




In this study, we systematically assessed the propagation characteristics of multiple drought types across global land areas. The SPI, SRI, and SSI time series at different accumulation periods from 1958 to 2024 were obtained by integrating the ERA5, GLDAS, and TerraClimate datasets, representing meteorological, hydrological, and agricultural drought conditions, respectively. Based on the correlation analysis and run theory, the response time, propagation rate, and lag time across meteorological, hydrological, and agricultural droughts were examined. Furthermore, the XGBoost-SHAP model was utilized to quantify the crucial feature factors that influence drought propagation. Main finding are: (1) The average response time (RT<sub>MH</sub> = 5.0 months, RT<sub>MA</sub> = 8.7 months, RT<sub>HA</sub> = 5.8 months), propagation rate (PR<sub>MH</sub> = 55.3%, PR<sub>MA</sub> = 30.3%, PR<sub>HA</sub> = 35.0%), and lag time (LT<sub>MH</sub> = 1.23 months, LT<sub>MA</sub> = 2.60 months, LT<sub>HA</sub> = 2.49 months) confirm the propagation pathway of meteorological-hydrological-agricultural drought at global scale. (2) Over the past six decades, approximately 40% of the landmass demonstrates temporal variations in the response time of drought propagation, while approximately 70% of the landmass shows temporal changes in both propagation rate and lag time. (3) Among the eight climatic, topographic, and vegetation-related feature factors, temperature and potential evapotranspiration are the primary factors influencing the propagation from meteorological drought to hydrological drought, primarily due to the lagging effects associated with snow melting processes. (4) In comparison, precipitation predominantly determines the propagation from meteorological or

hydrological drought to agricultural drought, due to the hydrological process effects of deep soil moisture and aquifer water. In summary, our study presents a multiple data-driven, global perspective on the propagation of meteorological, hydrological, and agricultural drought conditions, offering significant implications for drought monitoring and early warning systems in the context of global warming.

#### 455 Acknowledgements

This research was supported by the National Key R&D Program of China (2023YFC3205600).

## Data availability


ERA5 reanalysis dataset developed by the European Centre for Medium-Range Weather Forecasts (ECMWF) can be assessed at https://cds.climate.copernicus.eu/. GLDAS dataset is available at https://ldas.gsfc.nasa.gov/. TerraClimate dataset is downloaded from https://doi.pangaea.de/10.1594/PANGAEA.909132.

#### **Author contribution**

Conceptualization: YRL, Data curation: TH and YRL, Formal analysis: TH and YRL, Funding acquisition: LY, Investigation: TH and YRL, Methodology: YRL, Software: YRL, Validation: JWY and YRL, Visualization: YRL, Writing (original draft preparation): YRL, Writing (review and editing): JWY and LY.

# 465 Competing interests

The authors declare that they have no conflict of interest.

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
