# Peer review of "Understanding spatio-temporal patterns of the propagation characteristics across meteorological, hydrological, and agricultural droughts and their influencing factors"

_EGUsphere, 2025_

## Author Comment (AC1)

**RESPONSES TO REVIEWER ONE'S COMMENTS**

We are grateful to Reviewer #1 for his/her insightful review. The provided comments have contributed substantially to improving the paper. According to them, we have made significant efforts to revise the manuscript, with the details explained as follows:

**Point #1**

*COMMENT: Novelty. (a) Which is the specific research gap that this work addresses? This is currently not entirely clear from the title, abstract, and introduction. As the authors also acknowledge in the introduction, global-scale drought propagation studies are already available. I might see the originality of this work being the use of multiple datasets, but in this case, I believe this could be better worked out throughout the manuscript. Not only in the title, abstract, and introduction, to set the reader's expectations clear, but also later in the manuscript. (b) As a reader, for instance, I would enjoy having more discussion on the most suitable dataset(s) for drought applications. I see that providing a clear recommendation on this may be difficult from the current analyses, since you do not have here observations to benchmark the datasets with, but maybe you could still say something based on expectations on drought propagation that we have from previous observation-based studies? (c) Also, I would find useful to have clear ranges of variation for the drought propagation characteristics from the different datasets in the abstract and conclusions, as an indication of the uncertainties in such characteristics.*

*RESPONSE:* **(a)** We sincerely thank the reviewer for the essential comment on how to better articulate the novelty and contribution of our work. We agree that our initial presentation did not sufficiently highlight the distinctive research gap we are addressing. In the revised manuscript, we have thoroughly revised the Title, Abstract, and Introduction to precisely define our specific novelty and contribution, and have emphasized this message in the Discussion and Conclusion. In detail, the revised parts are provided as follows:

> **Title:** "Understanding meteorological, hydrological, and agricultural drought propagation and their influencing factors in an ensemble of multiple datasets" **(lines 1-3 of the revised manuscript)**

> **Abstract:** "Understanding the propagation of diverse drought conditions is necessary for drought preparedness. This study conducted a comprehensive evaluation of the propagation of meteorological, hydrological, and agricultural droughts across global land areas from 1958 to 2024, based on an ensemble of reanalysis data (ERA5), land surface model simulations (GLDAS), and merged observational datasets (TerraClimate). Two distinct methodological frameworks were employed to characterize drought propagation: time-lag correlation analysis and multi-threshold run theory. Based on standardized drought indices derived from precipitation (meteorological), runoff (hydrological) and soil moisture (agricultural), the drought propagation characteristics of response time (RT), propagation rate (PR), and lag time (LT) were examined. Moreover, the climatic and geographical factors influencing drought propagation were quantified using the SHapley

Additive exPlanations (SHAP)-based attribution method. The results demonstrate the propagation pathways of meteorological-hydrological-agricultural drought at the global-scale, with the average RT, PR, and LT from meteorological to hydrological drought at 5.0 months, 55.3%, and 1.23 months; from meteorological to agricultural drought at 8.7 months, 30.3%, and 2.60 months; and from hydrological to agricultural drought at 5.8 months, 35.0%, and 2.49 months, respectively. Notable temporal and spatial heterogeneities are observed in the drought propagation characteristics, which are closely influenced by with the regional climatic feature. Globally, temperature and potential evapotranspiration are the primary factors influencing the propagation of meteorological drought to hydrological drought, whereas precipitation plays a decisive role in the propagation from meteorological or hydrological drought to agricultural drought. The findings underscore the importance of taking climatic characteristics into account in the development and implementation of regional drought risk management." **(lines 9-26 of the revised manuscript)**

**Introduction:** "Drought is one of the most frequent natural disasters and is generally defined as a prolonged period of moisture deficits within the water cycle (Liu et al., 2020; AghaKouchak et al., 2023). Under global warming, the magnitude, frequency, and spatial extent of droughts have increased in recent decades, driven by precipitation variability and increased atmospheric evaporative demand (Chen et al., 2025; Gebrechorkos et al., 2025). The intensified droughts pose a significant threat to ecosystems and socio-economic sectors, such as agricultural production (Hendrawan et al., 2022), ecosystem productivity (Cao et al., 2022; Gu et al., 2025), and water resources (Fowler et al., 2022; Liu et al., 2025; Xie et al., 2025). Moreover, multiple climate and hydrological projections reveals a consistent drying trend across many regions during the 21st century (Cook et al., 2020; Christian et al., 2023; Li et al., 2025). The increasing frequency and intensity of droughts are expected to exacerbate water scarcity and ecosystem degradation in the foreseeable future, thus posing significant threats to both the natural environment and human society. Therefore, characterizing the spatio-temporal dynamics of droughts is of crucial importance, as it has significant implications for adapting to and mitigating the impacts of drought-related hazards.

Drought is a complex and multifaceted natural phenomenon (Wu et al., 2022). Although a drought event typically originates from inadequate precipitation and excessive evapotranspiration (referred to as meteorological drought), its impacts on human and natural systems are closely related to subsequent development, such as diminished runoff (hydrological drought), reduced soil moisture (agricultural drought), and declined groundwater (groundwater drought). There exists a strong interrelationship among different types of droughts, owing to the close linkage of their driving factors within the hydrological cycle. The transition processes from one type of drought (i.e., meteorological drought) to another (i.e., hydrological drought) are referred to as drought propagation (Apurv et al., 2017; Colombo et al., 2024). Understanding drought propagation characteristics, such as propagation time, probability, and threshold, are essential for elucidating drought occurrence and evolution mechanisms, which help facilitate the effective drought monitoring and early warning systems. Over the past decades, numerous studies have assessed the propagation characteristics of different drought conditions, using identification methods such as time-lag correlation analysis (López-Moreno et al., 2013; Barker et al., 2016; Liu et al., 2023; Geng et al., 2024), run theory (Aryal et al., 2024; Xiong et al., 2025), and event-coincidence analysis (Baez-Villanueva et al., 2024). For example, Shi et al. (2022a) examined the response time from meteorological and hydrological droughts using the maximum correlation

coefficient method, and analyzed the variations in response time across different climatic regions. Han et al. (2023) investigated the propagation pathways of various types of droughts across China and revealed the long-chain propagation mechanisms involving meteorological, hydrological, agricultural, and groundwater droughts. Aryal et al. (2024) evaluated the propagation time from meteorological drought to hydrological and agricultural drought across Australia, emphasizing the significant influence of climatic conditions and drought indices in assessing drought propagation dynamics. Among the aforementioned studies, correlation analysis and run theory are two of the most commonly used methodologies for quantifying drought propagation (Zhang et al., 2022). The time-lag correlation analysis based on standardized drought indices (SDIs) with varying accumulation periods provides a direct and effective method for assessing the response time among different drought types from a statistical perspective. However, the correlation analysis can only reflect the average linear relationship between different drought types, but cannot capture the variations among individual drought events (Zhou et al., 2024; Brunner and Chartier-Rescan, 2024). In comparison, the run theory identifies discrete drought events based on the time series of drought indices, thereby providing a more physically meaningful interpretation of the time lag relationship among various drought conditions. Although extensive research has been conducted to examine the characteristics of drought propagation, the results of these studies are heavily dependent on the datasets, evaluation methods, drought indices, and thresholds employed. A comparison of the differences and consistencies in drought propagation characteristics derived from different datasets and methods is desired to improve our understanding of drought propagation, particularly at the global scale.

The propagation characteristics of different types of drought vary depending on climatic conditions and underlying surface conditions (Apurv et al., 2017; Sattar et al., 2019; Apurv and Cai, 2020). Over the past decades, a large number of attribution studies have been conducted to quantify the impacts of climatic and geographical factors on drought propagation, using methods such as statistical analysis (Gevaert et al., 2018), clustering analysis (Liu et al., 2023), causality analysis (Shi et al., 2022b), and machine learning models (Muthuvel and Qin, 2025). Data-driven machine learning approaches are increasingly being employed in drought modeling due to their inherent advantages in capturing nonlinear patterns from complex and high-dimensional data (Sundararajan et al., 2021; Prodhan et al., 2022). Although machine learning models achieve satisfactory simulation accuracy, their reliability remains questionable due to their black-box nature and lack of physical interpretability (Rudin, 2019; Hassija et al., 2024). SHapley Additive exPlanations (SHAP) provides a unified attribution framework for explaining the machine learning outputs, enabling the interpretation of the causal relationships between independent variables and dependent variables (Antwarg et al., 2021; Nohara et al., 2022). Recently, SHAP-based attribution models have been utilized to investigate drought dynamics across various temporal and spatial scales (Xue et al., 2024; Feng et al., 2025). To the best of our knowledge, there is currently a lack of studies that analyze the factors influencing drought propagation from the perspective of SHAP-based machine learning.

Reliable drought monitoring and an in-depth understanding of the underlying mechanisms depend on datasets that accurately describe variations in drought-related hydro-meteorological variables. For large-scale and global assessments, gauge observations and gauge-based gridded datasets are often constrained by limited spatial and temporal coverage, the occurrence of missing values, and challenges in data accessibility (Wang et al., 2020; Gebrechorkos et al., 2024). Numerous satellites,

reanalysis, earth system models, and merged datasets have been developed, providing long-term and spatially continuous records of hydro-meteorological variables (Abatzoglou et al., 2018; Hersbach et al., 2020). Over the past decades, extensive efforts have been conducted to evaluate drought dynamics using different datasets at both regional and global scales. For example, Yuan et al. (2023) assessed the global patterns of flash drought, which is characterized by the rapid depletion of soil moisture, using ERA5 reanalysis and climate model datasets. Gebrechorkos et al. (2025) revealed the critical role of atmospheric evaporative demand in accelerating global drought severity, based on an ensemble of reanalysis, gridded observation, and hydrological model datasets. Wu et al. (2025) conducted an assessment of the dynamic predictability of agricultural drought across global land areas, utilizing the gridded observation, reanalysis, and hydrological model datasets. However, inconsistent findings across studies can be attributed to the inherent uncertainties and errors within different datasets; few systematic comparisons have been conducted to quantify the discrepancies among the multiple datasets in representing drought propagation characteristics (Chen et al., 2020; Huang et al., 2025).

Although previous studies have evaluated drought propagation across various temporal and spatial scales, a comprehensive assessment of the propagation characteristics of meteorological, hydrological, and agricultural droughts—derived from ensembles of multiple datasets—remains lacking, particularly at the global scale. Moreover, comparisons among different evaluation methods are also needed to fully understand the drought propagation process and its underlying influencing factors. Therefore, the objectives of this study are as follows: (1) to assess the spatial and temporal patterns of response time, propagation rate, and lag time across meteorological, hydrological, and agricultural droughts—derived from an ensemble of multiple datasets; (2) to quantify the main climatic and underlying surface factors that influence drought propagation characteristics from the perspective of machine learning; (3) to compare the robustness and uncertainty associated with different methods and datasets in characterizing drought propagation." **(lines 28-110 of the revised manuscript)**

**(b)** We agree that identifying the most appropriate dataset for drought applications is beneficial for the community, especially for the practical implementation of drought risk management. However, current research is unable to provide a single recommendation due to insufficient continuous observation in both time and space. In response the reviewer's comments, we have added a discussion on the relative merits and potential suitability of different datasets in characterizing drought propagation. Specifically, the revised paragraphs are provided as follows:

"A wide variety of meteorohydrological datasets are available for drought monitoring and evaluation. However, a consensus on the most suitable datasets for assessing drought propagation remains elusive across different applications and specific regions. Our results, derived from an ensemble of three different datasets (i.e., ERA5, GLDAS, and TerraClimate), reveal both robust global patterns and notable uncertainties in quantifying drought propagation characteristics. Consistent spatial patterns of drought propagation characteristics—such as shorter RT and LT in tropical and subtropical regions, and longer RT and LT in high-latitude and arid regions—across multiple datasets demonstrate the robustness of drought propagation mechanisms under climatic control. This agreement underscores the fundamental dynamics of drought propagation, which are independent of the methodology and forcing datasets. However, the magnitudes of drought propagation, especially in the meteorological to hydrological pathway, also demonstrate significant

inter-datasets variability (Figs. 2, 5, and 6). This divergence highlights the inherent uncertainty in drought propagation assessments and points to the distinct strengths and limitations of each dataset.

ERA5 is a high-resolution reanalysis dataset derived from the Integrated Forecasting System, which is forced by atmospheric observations. It generally exhibits higher values of $RT_{MH}$, $RT_{HA}$, $LT_{MH}$, and $LT_{HA}$, and lower values of $PR_{MH}$ and $PR_{HA}$ in high-latitude regions (Figs. S3-S6). This may more accurately represent the drought propagation in snow-dominated systems where runoff generation processes are complex and exhibit seasonal lags. GLDAS is an ensemble of multiple land surface models and exhibits intermediate drought propagation characteristics with relatively high spatial consistency in correlation coefficients (Fig. S4). This result indicates that the land surface model demonstrates a more robust pattern of drought propagation, and its process-consistent parameterizations may better represent the interrelationships among different drought types. TerraClimate, a statistically downscaled and bias-corrected dataset, consistently yielded the shortest $RT_{MH}$ and $LT_{MH}$ and the highest $PR_{MH}$, particularly in the mid- to low-latitudes. While this result aligns with the expectation of rapid response in rainfall-dominated regions, the empirical water balance model in TerraClimate may also lead to an overestimation of propagation speed and sensitivity. Our findings rely on the ensemble of multiple datasets, thus avoiding the bias of any single dataset and providing a more robust and consistent understanding of drought propagation." **(lines 489-512 of the revised manuscript)**

**(c)** We thank the reviewer for this crucial suggestion to enhance the transparency and robustness of our key findings. In the revised manuscript, we have added two metrics to evaluate drought propagation variability across different datasets: the coefficient of variation (CV) and the mean absolute deviation (MAD). The corresponding Results and Discussion section has also been revised, as provided below:

[Figure]

Figure 2. Spatial patterns of CV and MAD across the ERA5, GLDAS, and TerraClimate datasets for the response time from meteorological to hydrological droughts ($RT_{MH}$), from meteorological to agricultural droughts ($RT_{MA}$), and from hydrological to agricultural droughts ($RT_{HA}$). Larger values of the CV and MAD signify a more substantial disparity among distinct datasets.

"Moreover, the response time also varied across different datasets (Figs. S3 and S4). For example, consistently low $RT_{MH}$ values were observed in the TerraClimate dataset, whereas high $RT_{MH}$ values were observed in the ERA5 and GLDAS datasets. On average, the fluctuation ranges of $RT_{MH}$, $RT_{MA}$ and $RT_{HA}$ among different datasets were [1.96, 7.06] month, [7.87, 10.65] month, and [4.95, 8.00] month, respectively. To quantify the differences among various datasets, two metrics (i.e., CV and MAD) were calculated, and their spatial patterns are shown in Fig. 2. Larger values of CV and MAD indicate more substantial differences among different datasets. The lowest values of CV and MAD were observed in the $RT_{MA}$, followed by the $RT_{HA}$ and $RT_{MH}$. This result indicates that the $RT_{MA}$ showed relatively small variation across different datasets over global land areas, except in the high-latitude regions of the Northern Hemisphere. In contrast, substantial disparities existed in the $RT_{MH}$ and $RT_{HA}$ evaluations derived from different datasets, particularly in North America, the Sahara, central Asia, and central Australia. Specifically, consistently low $RT_{MH}$ values were observed in the TerraClimate dataset, whereas high $RT_{MH}$ values were predominantly found in the high latitudes of the Northern Hemisphere (Fig. S3). Regarding the maximum correlation of $RT_{MH}$, $RT_{MA}$, and $RT_{HA}$, the GLDAS dataset generally exhibited higher correlation coefficients, whereas the ERA5 and TerraClimate datasets displayed obviously spatial heterogeneity (Fig. S4). This indicates that the response time among different droughts is more reliably represented in the GLDAS dataset." **(lines 267-280 of the revised manuscript)**

[Figure]

**Figure 5.** Spatial patterns of CV and MAD across the ERA5, GLDAS, and TerraClimate datasets for the propagation rate from meteorological to hydrological droughts ($PR_{MH}$), from meteorological to agricultural droughts ($PR_{MA}$), and from hydrological to agricultural droughts ($PR_{HA}$).

[Figure]

**Figure 6.** Spatial patterns of CV and MAD across the ERA5, GLDAS, and TerraClimate datasets for the lag time from meteorological to hydrological droughts ($LT_{MH}$), from

meteorological to agricultural droughts ($LT_{MA}$), and from hydrological to agricultural droughts ($LT_{HA}$).

"Figs. 5 and 6 illustrate the spatial patterns of CV and MAD for propagation rate and lag time. The spatial patterns of propagation rate and lag time across the ERA5, GLDAS, and TerraClimate datasets are shown in Figs. S3 and S4. For different datasets, the average fluctuation ranges of $PR_{MH}$, $PR_{MA}$ and $PR_{HA}$ are [44.3, 72.8]%, [26.9, 33.6]%, and [24.9, 43.5]%, and those of $LT_{MH}$, $LT_{MA}$ and $LT_{HA}$ are [0.69, 1.49] month, [2.05, 3.03] month, and [1.87, 3.46] month, respectively. Consistent with the response time results, the $PR_{MA}$ and $LT_{MA}$ exhibited smallest differences across different datasets with low CV and MAD. In comparison, the relatively large differences were observed in the $PR_{MH}$, $PR_{HA}$, $LT_{MH}$, and $LT_{HA}$, especially in regions where in-situ observations are scarce. The highest $PR_{MH}$ and lowest $LT_{MH}$ values are found in the TerraClimate datasets, with $PR_{MH}$ values in the low and middle latitudes approaching 90% and $LT_{MH}$ values approaching less than 1 month, respectively." **(lines 331-339 of the revised manuscript)**

**Point #2**

*COMMENT: Agricultural drought definition and propagation from hydrological to agricultural droughts. The adopted agricultural drought definition and the choice of investigating hydrological-to-agricultural drought propagation is not entirely clear to me. Agricultural droughts are introduced in the paper as 'reduced soil moisture' (L40), coherently with extensive previous literature (Van Loon, 2015) which refers to agricultural or soil moisture droughts as deficits in the root-zone soil moisture mainly impacting the agricultural sector, following meteorological droughts and potentially leading to hydrological droughts (i.e., deficits in runoff and groundwater, Van Loon, 2015). Previous works therefore mostly investigated the propagation from meteorological to agricultural and then to hydrological droughts, by finding shorter propagation times from meteorological to agricultural droughts than from meteorological to hydrological droughts (e.g., Odongo et al., 2023; Teutschbein et al., 2025). Could you clarify why you chose to investigate the propagation from hydrological to agricultural droughts instead of the other way round? Also, are you considering soil moisture data from the upper or deeper layers? This is not specified in the methods. In the discussion, agricultural droughts are said to affect the 'deeper soil moisture' (L393), which may point to the use of deep soil moisture data only, but I ask you to clarify earlier on this important piece of information for the general understanding of the work. The use of deep soil moisture would (partly) explain to me both the choice of investigating this drought propagation pathway and the results, showing longer propagation times for agricultural than for hydrological droughts with respect to the meteorological ones. Yet, if this is the case, I wonder whether the use of the term agricultural droughts is the most suitable here, given the interest in the upper soil layer by the agricultural sector.*

*RESPONSE:* We sincerely appreciate the reviewer's insightful comments. **(a) Agricultural drought definition.** In this study, we considered the soil moisture in the root zone layers within a depth of 1 meter to define the agricultural drought. The soil moisture data were derived from the ensemble of ERA5, GLDAS, and TerraClimate datasets, which have different soil layer structures. Thus, we aggregated volumetric soil water content to a 1-meter depth using weighted data from different soil layers. For example, the soil moisture in ERA5 was aggregated to 1

meter volumetric soil water using weighted data from three layers: 0–7 cm, 7–28 cm, and 28–100 cm. In the updated versions, we have revised the sections "2.1 Datasets" and "2.2 Drought definition and drought indices" to clarify the details of the soil moisture data utilized and the definition of agricultural drought. In addition, the description of the drought propagation mechanism in the discussion section has been rephrased as well. The revised paragraphs are provided as follows:

"2.1 Datasets
Monthly precipitation, runoff, and soil moisture were derived from the ERA5, the Global Land Data Assimilation System (GLDAS), and TerraClimate datasets to calculate the drought indices. ERA5 is the fifth-generation global atmospheric reanalysis product developed by the European Centre for Medium-Range Weather Forecasts. It integrates extensive records of both in-situ and satellite observations through an ensemble-based data assimilation system (Hersbach et al., 2020). Precipitation in ERA5 was generated by the atmospheric component of the Integrated Forecasting System, whereas runoff and soil moisture were simulated by a land surface model (Boussetta et al., 2021). The soil moisture in ERA5 was aggregated to 1 meter volumetric soil water using weighted data from three layers: 0–7 cm, 7–28 cm, and 28–100 cm. GLDAS is a multi-model ensemble comprising three land surface models—Noah, Catchment, and the Variable Infiltration Capacity—which integrate satellite and in-situ observations through advanced land surface modeling techniques. The soil moisture in GLDAS models has different soil layer structures, all of which were weighted to the root zone depth of 1 meter to be consistent with ERA5. TerraClimate integrates multiple datasets, including WorldClim, Climate Research Unit, and Japanese 55-year Reanalysis, to generate hydro-meteorological variables (Abatzoglou et al., 2018). The soil moisture in the TerraClimate refers to the plant extractable soil water based on the root zone storage capacity, as modeled by an empirical water balance model. To ensure spatial and temporal consistency, the period from 1958 to 2024 was selected as the reference period, and all datasets were uniformly interpolated onto a 1 °×1 °latitude–longitude grid using bilinear interpolation.

In addition, the temperature and potential evapotranspiration (PET) were also obtained from the ensemble of ERA5, GLDAS, and TerraClimate datasets. Potential evapotranspiration in these datasets was calculated using the Penman-Monteith method (Abatzoglou et al., 2018). The Normalized Difference Vegetation Index (NDVI) was obtained directly from the Advanced Very High Resolution Radiometer instruments operated by the National Oceanic and Atmospheric Administration (NOAA) (Pinzon and Tucker, 2014). The elevation dataset was obtained from the ETOPO Global Relief Model developed by the National Centers for Environmental Information (https://www.ncei.noaa.gov/products/etopo-global-relief-model). The aridity index dataset was derived from the Global Aridity Index and Potential Evapotranspiration Database—Version 3 (Zomer et al., 2022)." **(lines 112-136 of the revised manuscript)**

"2.2 Drought definition and drought indices
Drought is a complex phenomenon related to multiple variables in the water cycle, and there is no universally accepted definition in the current literature (Van Loon, 2015). Drought propagation is a hierarchical top-down process in which meteorological drought, caused by insufficient precipitation, can propagate to other hydrological variables (Teutschbein et al., 2025). A large number of drought indices and datasets have been developed to characterize varying drought conditions at different spatial and temporal

scales (AghaKouchak et al., 2023; Gebrechorkos et al., 2025). To provide a consistent and comparable assessment of drought propagation, standardized drought indices (SDIs) derived from precipitation, runoff, and soil moisture were used to define meteorological, hydrological, and agricultural droughts. The SDI time series were obtained by fitting the drought variables of interest to a suitable probability distribution and subsequently normalizing the probabilities to generate a standardized time series. Three SDIs, including Standardized Precipitation Index (SPI) (McKee et al., 1993), Standardized Runoff Index (SRI) (Shukla and Wood, 2008), and Standardized Soil Moisture Index (SSI) (Hao and AghaKouchak, 2013), were computed by fitting parametric probability distributions, specifically the Gamma, log-normal, and normal distributions, respectively. The maximum likelihood estimation (MLE) method was utilized to estimate the parameters of the probability distributions, with the initial values determined based on the L-moments estimation (Stagge et al., 2015). Compared with other drought indices, the SDI is not only simple and efficient to calculate, but also applicable to diverse climates due to its consistent calculation procedure (Zarch et al., 2015; Adnan et al., 2018). Meanwhile, SDI can be calculated using drought-related variables across multiple time scales, thus enhancing its effectiveness in analyzing drought propagation." **(lines 137-154 of the revised manuscript)**

"4.1. Physical interpretation of drought propagation characteristics
In this study, two distinct methodological frameworks were employed to quantify drought propagation: (1) the response time derived from time-lag correlation analysis, and (2) the lag time based on event identification using the run theory. Response time is determined by identifying the accumulation period of a drought index (e.g., SPI) that maximizes its correlation with a target drought index (e.g., SSI at a 1-month accumulation timescale) (López-Moreno et al., 2013; Zhang et al., 2022). This approach reflects the overall synchronicity and statistical memory characteristics of various drought conditions. Thus, the response time values are strongly influenced by long-term variations in regional climatic and hydrological conditions, such as the seasonal cycle, multi-year climate oscillations, and water storage capacity. The response time refers to the system's long-term state that retains a memory of past drought conditions. The evaluation of response time is beneficial for seasonal drought predictability and long-term drought preparedness. The response time also functions as an indicator of the feasibility of using one type of drought index as a proxy for another. For example, due to the lack of comprehensive observational data, the SPI with varying accumulation periods can reflect hydrological, agricultural and groundwater drought conditions (Kumar et al., 2016).

In comparison, lag time is derived from discrete drought events identified using the multi-threshold run theory, which measures the time difference between the onset of one drought event and the onset of another drought event. By focusing on event-based dynamics, the lag time reflects the instantaneous triggering mechanism by which drought signals propagate from the atmosphere to the land surface. Numerous previous studies have analyzed the threshold of extreme stress that triggers drought propagation, using methods such as copula functions, hydrological models, and machine learning (Geng et al., 2024; Yang et al., 2025). The lag time captures the non-linear response mechanism between different drought conditions at a short time scale, which is crucial for real-time early warning and impact assessment.

Our results provide a globally consistent comparison of the response time and lag time for meteorological, hydrological, and agricultural drought propagation. The response

time of drought propagation (average $RT_{MH}$, $RT_{MA}$, and $RT_{HA}$ of 5.0 [2.7, 6.7] months, 8.7 [5.0, 11.3] months, and 5.8 [2.3, 7.3] months) is generally longer than the lag time (average $LT_{MH}$, $LT_{MA}$, and $LT_{HA}$ of 1.23 [0.68, 1.68] months, 2.60 [1.71, 2.92] months, and 2.49 [1.68, 2.51] months). This numerical gap arises from differences in the methodology, but both approaches indicate a consistent propagation pathway for meteorological, hydrological, and agricultural droughts, with similar spatial patterns. In addition, the machine learning-based attribution method also identifies similar impact factors, which indicates the consistency of drought propagation mechanisms revealed by response time and lag time. This aligns with the conceptual framework of drought propagation, where precipitation deficits (meteorological drought) first influence runoff generation over the land surface (hydrological drought), and subsequently affect soil moisture in the root zone (agricultural drought)." **(lines 435-466 of the revised manuscript)**

**(b) Propagation from hydrological to agricultural droughts.** As described in the references of Van Loon (2015) and Teutschbein et al. (2025), drought propagation is generally regarded as a hierarchical top-down process. Meteorological drought, primarily caused by precipitation deficits, can cascade to other hydrological variables in the water cycle. From this perspective, numerous studies have evaluated the propagation of meteorological drought to other types of drought. Hydrological drought is a broad term referring to negative anomalies in surface and subsurface water, including groundwater levels, lake water levels, and river discharge. When hydrological drought is defined by streamflow, the pathway of drought propagation is from meteorological to agricultural and then to hydrological droughts (Teutschbein et al., 2025). In comparison, when hydrological drought is defined by runoff, it shows that hydrological drought propagates to agricultural drought (Han et al., 2023), as shown in Figure 3 in the reference of Van Loon (2015). In our analysis, we characterized meteorological, hydrological, and agricultural droughts based on standardized drought indices (SDIs) derived from precipitation, runoff, and soil moisture. These hydrological variables were directly obtained from three different datasets (i.e., ERA5, GLDAS, and TerraClimate), where runoff is the volume of water that originates from precipitation and flows over the land surface. It is not directly equal to the streamflow in the stream channels. The utilization of runoff to represent hydrological drought mainly stems from the fact that our study focuses on the propagation of droughts at the global scale. The runoff exhibits a substantial advantage regarding data availability, featuring continuity in both temporal and spatial dimensions. In the revised manuscript, we have revised the "2.2 Drought definition and drought indices" section of the methodology to elucidate the drought indices employed in this study. In addition, we have incorporated comparisons with other studies in the discussion section to clarify the reasons for the study's focus on the propagation from hydrological to agricultural droughts and to emphasize the significance of drought indices in understanding drought propagation. In detail, the revised paragraphs are provided as follows:

"Drought propagation evaluation relies heavily on drought indices for monitoring and characterizing various drought types. Considering the data availability and the continuity in both temporal and spatial dimensions at the global scale, we employed the SPI, SRI, and SSI to represent meteorological, hydrological, and agricultural droughts. Our results demonstrated the propagation pathway of meteorological-hydrological-agricultural droughts, which is consistent with previous studies that employed similar indices (Han et al., 2023; Mei et al., 2025). As a multifaceted phenomenon, hydrological drought is a broad term that is related not only to runoff but also to streamflow and the levels of

groundwater, lakes, and reservoirs (Van Loon, 2015). Using the drought indices derived from streamflow, the propagation from agricultural to hydrological droughts has also been identified in many studies, particularly at the watershed scale (Odongo et al., 2023; Teutschbein et al., 2025). Runoff is the volume of water that originates from precipitation and flows over the land surface; it is not directly equal to the streamflow in stream channels. A deficit in runoff can directly affect the availability of soil moisture due to reduced recharge to the root zone, representing the propagation from hydrological drought to agricultural drought. In comparison, soil moisture retains precipitation that falls on the land surface and then delays the propagation time form precipitation to streamflow (McColl et al., 2017)." **(lines 512-524 of the revised manuscript)**

**Point #3**

*COMMENT: **Trend analysis.** **(a)** Important methodological details on this are missing. The analysis is briefly described in the Results section (L251–252), but how this moving-window trend analysis exactly works is not totally clear. From my understanding, you calculate the various metrics (e.g., propagation time) for each year based on a moving window consisting of (the next?) 30 years and then apply a trend analysis on the annual values that you obtained. If this is the case, how do you deal with potential autocorrelation from partially overlapping raw data? I would suggest expanding on this point and moving the current description of this analysis to the Methods section (e.g., to a new subsection between the current 2.4 and 2.5). Please also provide full name and appropriate references for the statistical tests used here (i.e., the 'M-K test' currently mentioned in the text). **(b)** Finally, what do you mean by 'monotonic trend' in the pie charts in e.g. Fig. 3? From my understanding, it refers to the greyish areas in the maps, with trend slopes close to zero. Did you set any lower and upper limits on the trend slopes to discriminate these 'monotonic trends'? If so, please specify. **(c)** And why are these monotonic trends not appearing in the pie charts in Fig. 5 and 6, even though greyish areas are reported in the corresponding maps?*

*RESPONSE:* We sincerely appreciate the reviewer's insightful comment. **(a)** We acknowledge that the description of the moving-window trend analysis in the original manuscript was insufficient. In this study, we employed a moving window approach to evaluate the temporal pattern of drought propagation characteristics. For each grid, the response time, propagation rate, and lag time between different types of droughts were calculated using a 30-year moving window that advanced one year at a time, thereby generating an annual time series for the period from 1987 to 2024. To avoid the potential autocorrelation from overlapping data, we conducted the Trend-Free Pre-Whitening (TFPW) procedure prior to the MK test to address potential autocorrelation (Yue et al., 2002). The TFPW-MK test effectively removes the influence of serial correlation on trend significance, providing a more robust assessment. In the revised manuscript, we have added a new subsection, "2.6 Trend analysis of drought propagation," to make these critical methodological details clear. In detail, the revised paragraphs are provided as follows:

"2.6 Trend analysis of drought propagation
Temporal evolution of drought propagation characteristics was assessed through a moving window approach. For each grid, the drought propagation characteristics (i.e., response time, propagation rate, and lag time) between different types of droughts were calculated using a 30-year moving window that advanced one year at a time, thereby

generating an annual time series for the period from 1987 to 2024. The significance of the time series trend was assessed using the Mann-Kendall (MK) test, where a trend was considered statistically significant at the p-value < 0.05 (Mann, 1945; Kendall, 1975). Given that the series is derived from a moving window with overlapping data, we conducted the Trend-Free Pre-Whitening (TFPW) procedure prior to the MK test to address potential autocorrelation (Yue et al., 2002). The TFPW-MK test effectively removes the influence of serial correlation on trend significance, providing a more robust assessment. In addition, the magnitude of the trend was estimated using Sen's slope estimator (Sen, 1968)." **(lines 201-210 of the revised manuscript)**

**(b)** In this study, a monotonic trend is defined as the situation where Sen's slope is equal to 0. Accordingly, we have incorporated an explanation of the monotonic trend in Figure 3 to enhance its clarity. In detail, the revised parts are provided as follows:

[Figure]

Figure 3. Spatial patterns of time series trends in $RT_{MH}$, $RT_{MA}$ and $RT_{HA}$ across global land areas. The blank grids signify that, within at least one time-window in the time series of response time obtained from the moving window, the correlation coefficient is not statistically significant. The black dots indicate the statistical significance of the time series trend, where the p-value of the TFPW-MK test is less than 0.05. A significant increase (decrease) indicates that the Sen's slope is greater (less) than 0 and that the p-value of the TFPW-MK test is less than 0.05. A nonsignificant increase (decrease) indicates that the Sen's slope is greater (less) than 0 and that the p-value of the TFPW-MK test is greater than 0.05. A monotonic trend indicates that Sen's slope is equal to 0.

**(c)** Figs. 5 and 6 represent the spatial patterns of time series trends for the propagation rate and lag time, which exhibit a temporal pattern that is entirely inconsistent with that of the lag time (as

shown in Fig. 3). Although there are gray areas on the heat map, its Sen's slope is not exactly equal to 0, so it was not displayed on the pie chart.

**Point #4**

*COMMENT: **Language and readability.** I find the paper generally well structured, but rather lengthy and sometimes convoluted. I think the reading flow could be improved by reducing redundant expressions (e.g., couldn't 'feature factors' be simply 'features' or 'factors'?), repetitions between sections (e.g., L105–107 already said in the previous section), and rather obvious statements (e.g., 'with positive correlation with rP > 0, and negative correlation with rP < 0', L156–157). I also noticed many abbreviations, especially in Sect. 2.1 Datasets (e.g., ECMWF, CLSM, etc), which seem to me not used anymore in the paper. I would suggest removing them and making sure that abbreviations are always introduced the first time they are used (currently not the case, see e.g., ML at L73). Consistent notation throughout the text and across the text and the figures (currently not the case, see e.g., Eq.1 and Fig. 1b, d, and f) would also ease the readability of the paper. In summary, I see room for improvement, with another careful round of proofreading focused on language.*

*RESPONSE:* We sincerely thank the reviewer for the thorough and constructive feedback on the language, clarity, and presentation of our manuscript. We agree that the manuscript could be more concise and polished to improve readability. Accordingly, we have performed a line-by-line edit to eliminate redundant expressions in the revised manuscript. Examples corrected include:
- Changed the "feature factors" to "factors"
- Removed the "with positive correlation with rP > 0, and negative correlation with rP < 0"
- Delete the repetition sentences in the Dataset section
- Removed the unnecessary abbreviations in Section 2.1 Datasets
- Checked all the abbreviations in the manuscript (e.g., ML)
- Checked consistent notation throughout the text and figures

**Point #5**

*COMMENT: **References.** I appreciate the referencing to very recent literature on the topic, yet I believe that also additional references to (older) seminal papers on droughts and drought propagation characteristics would be appropriate (e.g., López-Moreno et al., 2013 and Barker et al., 2016 for the correlation analysis, other papers that I referred to above).*

*RESPONSE:* We sincerely appreciate the reviewer's helpful comment. We agree that citing seminal works is crucial for properly contextualizing our study within the historical development of the field and for acknowledging foundational concepts. According to the reviewer's comments, we have incorporated these references into the revised manuscript. Specifically, the revised sentences and the corresponding references are provided below:

"Over the past decades, numerous studies have assessed the propagation characteristics of different drought conditions, using identification methods such as time-lag correlation

analysis (López-Moreno et al., 2013; Barker et al., 2016; Liu et al., 2023; Geng et al., 2024), run theory (Aryal et al., 2024; Xiong et al., 2025), and event-coincidence analysis (Baez-Villanueva et al., 2024)." **(lines 49-52 of the revised manuscript)**

**The added references:**

López-Moreno, J. I., Vicente-Serrano, S. M., Zabalza, J., Beguería, S., Lorenzo-Lacruz, J., Azorin-Molina, C., and Morán-Tejeda, E.: Hydrological response to climate variability at different time scales: A study in the Ebro basin, J. Hydrol., 477, 175–188, doi:10.1016/j.jhydrol.2012.11.028, 2013.
Barker, L. J., Hannaford, J., Chiverton, A., and Svensson, C.: From meteorological to hydrological drought using standardised indicators, Hydrol. Earth Syst. Sc., 20, 2483–2505, doi:10.5194/hess-20-2483-2016, 2016.

**Point #6**

*COMMENT: Abstract, I would appreciate introducing the datasets and methods you used in general terms (e.g., reanalyses for ERA5 and so on), to facilitate readers potentially not familiar with these specific datasets and methods.*

*RESPONSE:* We sincerely appreciate the reviewer's insightful comment. We agree that describing the datasets and methods in more general terms will significantly improve the accessibility of our manuscript. In response, we have revised the description of the datasets and methods in the abstract to make it clearer. In detail, the revised sentences are provided as follows:

**Abstract:** "Understanding the propagation of diverse drought conditions is necessary for drought preparedness. This study conducted a comprehensive evaluation of the propagation of meteorological, hydrological, and agricultural droughts across global land areas from 1958 to 2024, based on an ensemble of reanalysis data (ERA5), land surface model simulations (GLDAS), and merged observational datasets (TerraClimate). Two distinct methodological frameworks were employed to characterize drought propagation: time-lag correlation analysis and multi-threshold run theory. Based on standardized drought indices derived from precipitation (meteorological), runoff (hydrological) and soil moisture (agricultural), the drought propagation characteristics of response time (RT), propagation rate (PR), and lag time (LT) were examined. Moreover, the climatic and geographical factors influencing drought propagation were quantified using the SHapley Additive exPlanations (SHAP)-based attribution method. The results demonstrate the propagation pathways of meteorological-hydrological-agricultural drought at the global-scale, with the average RT, PR, and LT from meteorological to hydrological drought at 5.0 months, 55.3%, and 1.23 months; from meteorological to agricultural drought at 8.7 months, 30.3%, and 2.60 months; and from hydrological to agricultural drought at 5.8 months, 35.0%, and 2.49 months, respectively. Notable temporal and spatial heterogeneities are observed in the drought propagation characteristics, which are closely influenced by with the regional climatic feature. Globally, temperature and potential evapotranspiration are the primary factors influencing the propagation of meteorological drought to hydrological drought, whereas precipitation plays a decisive role in the propagation from meteorological or hydrological drought to agricultural drought. The findings underscore the importance of taking climatic characteristics into

account in the development and implementation of regional drought risk management."
**(lines 9-26 of the revised manuscript)**

**Point #7**

*COMMENT: L26–28, I suggest rephrasing since, from my understanding, Gebrechorkos et al. (2025) showed that increases in atmospheric evaporative demand significantly contributed to recent increases in drought severity, but not as primary factor.*

*RESPONSE:* We sincerely appreciate the reviewer's helpful comment. Upon re-examining the cited study, we agree that Gebrechorkos et al. (2025) emphasized the significant role of increased atmospheric evaporative demand in recent drought severity, while not identifying it as the primary factor. Accordingly, we have revised the sentence to accurately reflect the role of increased atmospheric evaporative demand in the recent increase in drought severity. In the updated manuscript, the revised sentence is provided as follows:

> "Under global warming, the magnitude, frequency, and spatial extent of droughts have increased in recent decades, driven by precipitation variability and increased atmospheric evaporative demand (Chen et al., 2025; Gebrechorkos et al., 2025)." **(lines 29-31 of the revised manuscript)**

**Point #8**

*COMMENT: L42–43, I believe defining here drought propagation characteristics would be beneficial for readers who may not be familiar with them and to ease the readability of the rest of the manuscript.*

*RESPONSE:* We sincerely appreciate the reviewer's helpful suggestion. As recommended, we have revised this sentence to offer supplementary clarifications regarding the definition of the drought propagation characteristics. In detail, the revised sentence is provided as follows:

> "Understanding drought propagation characteristics, such as propagation time, probability, and threshold, are essential for elucidating drought occurrence and evolution mechanisms, which help facilitate the effective drought monitoring and early warning systems." **(lines 46-49 of the revised manuscript)**

**Point #9**

*COMMENT: L44–48, many different methods are currently mixed together in this sentence and specifically: methods used to generate the datasets needed for drought propagation studies (e.g., hydrological models), methods used to quantify drought propagation characteristics (e.g., correlation analysis and run theory), and methods used to attribute these characteristics to their controls (e.g., ML). I suggest clarifying this point, for instance by splitting this long sentence into several ones. In addition, maybe add event-coincidence analysis as another method to quantify*

*drought propagation characteristics as proposed by Baez-Villanueva et al. (2024)? Finally, I would suggest removing the mention to the complex network theory since this is used for spatial drought propagation, which is not the topic of this paper, or alternatively, specifying this point and what spatial drought propagation is.*

*RESPONSE:* We sincerely appreciate the reviewer's insightful comment. Accordingly, in the revised manuscript, we have rewritten this sentence to emphasize the methods for identifying drought propagation characteristics. The sentences and references related to datasets and attribution methods have been integrated into other paragraphs. In addition, we have added the method of event-coincidence analysis and removed the complex network theory. The revised sentence is provided as follows:

> "Over the past decades, numerous studies have assessed the propagation characteristics of different drought conditions, using identification methods such as time-lag correlation analysis (López-Moreno et al., 2013; Barker et al., 2016; Liu et al., 2023; Geng et al., 2024), run theory (Aryal et al., 2024; Xiong et al., 2025), and event-coincidence analysis (Baez-Villanueva et al., 2024)." **(lines 49-52 of the revised manuscript)**

**Point #10**

*COMMENT:* *L52, I suggest introducing the concept of groundwater droughts earlier.*

*RESPONSE:* We sincerely appreciate the reviewer's helpful comment. Accordingly, we have incorporated a concise description of the concept of groundwater droughts in the earlier sentence. The revised sentence is provided as follows:

> "Although a drought event typically originates from inadequate precipitation and excessive evapotranspiration (referred to as meteorological drought), its impacts on human and natural systems are closely related to subsequent development, such as diminished runoff (hydrological drought), reduced soil moisture (agricultural drought), and declined groundwater (groundwater drought)." **(lines 40-43 of the revised manuscript)**

**Point #11**

*COMMENT:* *L64–65, I would say that all the global-scale analyses cited before are 'consistent' and 'comparable' within themselves since they use common methods and datasets for the whole globe. I would suggest rephrasing this sentence to the exact research gap you are aiming at addressing with your work (see also comment #1).*

*RESPONSE:* We sincerely appreciate the reviewer's insightful comment. The original sentence was imprecise and did not effectively emphasize the research gap. Following the reviewer's comment #1, we have summarized the research gap regarding the differences and consistencies in drought propagation characteristics derived from different datasets and methods. Accordingly, the revised sentence is provided as follows:

"A comparison of the differences and consistencies in drought propagation characteristics derived from different datasets and methods is desired to improve our understanding of drought propagation, particularly at the global scale." **(lines 67-69 of the revised manuscript)**

**Point #12**

*COMMENT: L69–70, I do not fully agree with this sentence, which seems to me also contradicting the previous one. Literature on the factors controlling drought propagation across different climatic and geographical regions is rather vast now (see e.g., Xiong et al. 2025 and other reviews on the topic, also cited in the text).*

*RESPONSE:* We sincerely appreciate the reviewer's insightful comment. We agree with your observation that the literature on the factors controlling drought propagation is extensive and has grown significantly. Our study focused on identifying the factors influencing drought propagation from the perspective of a SHAP-based attribution approach. We acknowledge that the original phrasing was ambiguous and could be perceived as contradictory. Accordingly, we have revised the corresponding sentences to eliminate the contradictory description. The revised sentences are as follows:

"Over the past decades, a large number of attribution studies have been conducted to quantify the impacts of climatic and geographical factors on drought propagation, using methods such as statistical analysis (Gevaert et al., 2018), clustering analysis (Liu et al., 2023), causality analysis (Shi et al., 2022b), and machine learning models (Muthuvel and Qin, 2025)." **(lines 72-75 of the revised manuscript)**

**Point #13**

*COMMENT: L93, could you provide some references of these comparisons?*

*RESPONSE:* We sincerely appreciate the reviewer's helpful comment. Accordingly, we have added references comparing different datasets in characterizing the processes of drought propagation. In the updating manuscript, the revised sentence is provided as follows:

"However, inconsistent findings across studies can be attributed to the inherent uncertainties and errors within different datasets; few systematic comparisons have been conducted to quantify the discrepancies among the multiple datasets in representing drought propagation characteristics (Chen et al., 2020; Huang et al., 2025)." **(lines 98-100 of the revised manuscript)**

**The added references:**

Chen, N., Li, R., Zhang, X., Yang, C., Wang, X., Zeng, L., and Niyogi, D.: Drought propagation in Northern China Plain: A comparative analysis of GLDAS and MERRA-2 datasets, J. Hydrol., 588, 125026, doi:10.1016/j.jhydrol.2020.125026, 2020.
Huang, K., Zhang, H., Cui, G., Wang, Y., Yin, M., and Du, J.: Drought propagation in

china: Uncertainties originate more from dataset choice than drought index selection, Atmos. Res., 308, 108555, doi:10.1016/j.atmosres.2024.108555, 2025.

**Point #14**

*COMMENT:* *L105, I would argue that data quality is crucial for any study, not only drought studies. It may also be a matter of personal taste, but I do not see as really needed these very general sentences, which also contribute to making the paper quite lengthy in my view (see also comment #4).*

*RESPONSE:* We sincerely appreciate the reviewer's useful comment. We agree that data quality is a universal concern across all scientific studies; however, the previous sentence is a general statement and is therefore unnecessary in the "Data and Methodology" section. In response, we have revised the paragraphs regarding the datasets, and the revised paragraphs are provided below:

"2.1 Datasets
Monthly precipitation, runoff, and soil moisture were derived from the ERA5, the Global Land Data Assimilation System (GLDAS), and TerraClimate datasets to calculate the drought indices. ERA5 is the fifth-generation global atmospheric reanalysis product developed by the European Centre for Medium-Range Weather Forecasts. It integrates extensive records of both in-situ and satellite observations through an ensemble-based data assimilation system (Hersbach et al., 2020). Precipitation in ERA5 was generated by the atmospheric component of the Integrated Forecasting System, whereas runoff and soil moisture were simulated by a land surface model (Boussetta et al., 2021). The soil moisture in ERA5 was aggregated to 1 meter volumetric soil water using weighted data from three layers: 0–7 cm, 7–28 cm, and 28–100 cm. GLDAS is a multi-model ensemble comprising three land surface models—Noah, Catchment, and the Variable Infiltration Capacity—which integrate satellite and in-situ observations through advanced land surface modeling techniques. The soil moisture in GLDAS models has different soil layer structures, all of which were weighted to the root zone depth of 1 meter to be consistent with ERA5. TerraClimate integrates multiple datasets, including WorldClim, Climate Research Unit, and Japanese 55-year Reanalysis, to generate hydro-meteorological variables (Abatzoglou et al., 2018). The soil moisture in the TerraClimate refers to the plant extractable soil water based on the root zone storage capacity, as modeled by an empirical water balance model. To ensure spatial and temporal consistency, the period from 1958 to 2024 was selected as the reference period, and all datasets were uniformly interpolated onto a $1\,°×1\,°$ latitude–longitude grid using bilinear interpolation.

In addition, the temperature and potential evapotranspiration (PET) were also obtained from the ensemble of ERA5, GLDAS, and TerraClimate datasets. Potential evapotranspiration in these datasets was calculated using the Penman-Monteith method (Abatzoglou et al., 2018). The Normalized Difference Vegetation Index (NDVI) was obtained directly from the Advanced Very High Resolution Radiometer instruments operated by the National Oceanic and Atmospheric Administration (NOAA) (Pinzon and Tucker, 2014). The elevation dataset was obtained from the ETOPO Global Relief Model developed by the National Centers for Environmental Information

(https://www.ncei.noaa.gov/products/etopo-global-relief-model). The aridity index dataset was derived from the Global Aridity Index and Potential Evapotranspiration Database—Version 3 (Zomer et al., 2022)." **(lines 112-136 of the revised manuscript)**

**Point #15**

**COMMENT:** *L118 'high-spatial-resolution', explicitly mentioning the resolution of this and all other datasets would help the readers in my opinion.*

**RESPONSE:** We sincerely appreciate the reviewer's useful comment. In fact, the ERA5, GLDAS, and TerraClimate datasets have different temporal and spatial resolutions. To ensure spatial and temporal consistency, the period from 1958 to 2024 was selected as the reference period, and all datasets were uniformly interpolated onto a 1 °×1 °latitude–longitude grid using bilinear interpolation. In the updating manuscript, we have revised the sentences to make it clearer. The revised sentences are provided as follows:

> "To ensure spatial and temporal consistency, the period from 1958 to 2024 was selected as the reference period, and all datasets were uniformly interpolated onto a 1 °×1 ° latitude–longitude grid using bilinear interpolation." **(lines 126-127 of the revised manuscript)**

**Point #16**

**COMMENT:** *L121–123, I appreciate the details provided here on how potential evapotranspiration, runoff and soil moisture are computed in this dataset. Could you provide such details also for the other datasets? I think they would give to the reader more context, also on the differences you detect among them.*

**RESPONSE:** We sincerely appreciate the reviewer's helpful comment. Potential evapotranspiration in these datasets was calculated using the Penman–Monteith method; runoff and soil moisture were simulated by the land surface models in ERA5 and GLDAS, while in TerraClimate they were simulated by an empirical water balance model. Accordingly, we have revised the sentences to provide more detailed information about the datasets. The revised paragraphs are provided below:

> "2.1 Datasets
> Monthly precipitation, runoff, and soil moisture were derived from the ERA5, the Global Land Data Assimilation System (GLDAS), and TerraClimate datasets to calculate the drought indices. ERA5 is the fifth-generation global atmospheric reanalysis product developed by the European Centre for Medium-Range Weather Forecasts. It integrates extensive records of both in-situ and satellite observations through an ensemble-based data assimilation system (Hersbach et al., 2020). Precipitation in ERA5 was generated by the atmospheric component of the Integrated Forecasting System, whereas runoff and soil moisture were simulated by a land surface model (Boussetta et al., 2021). The soil moisture in ERA5 was aggregated to 1 meter volumetric soil water using weighted data from three layers: 0–7 cm, 7–28 cm, and 28–100 cm. GLDAS is a multi-model ensemble

comprising three land surface models—Noah, Catchment, and the Variable Infiltration Capacity—which integrate satellite and in-situ observations through advanced land surface modeling techniques. The soil moisture in GLDAS models has different soil layer structures, all of which were weighted to the root zone depth of 1 meter to be consistent with ERA5. TerraClimate integrates multiple datasets, including WorldClim, Climate Research Unit, and Japanese 55-year Reanalysis, to generate hydro-meteorological variables (Abatzoglou et al., 2018). The soil moisture in the TerraClimate refers to the plant extractable soil water based on the root zone storage capacity, as modeled by an empirical water balance model. To ensure spatial and temporal consistency, the period from 1958 to 2024 was selected as the reference period, and all datasets were uniformly interpolated onto a 1 °×1 °latitude–longitude grid using bilinear interpolation.

In addition, the temperature and potential evapotranspiration (PET) were also obtained from the ensemble of ERA5, GLDAS, and TerraClimate datasets. Potential evapotranspiration in these datasets was calculated using the Penman-Monteith method (Abatzoglou et al., 2018). The Normalized Difference Vegetation Index (NDVI) was obtained directly from the Advanced Very High Resolution Radiometer instruments operated by the National Oceanic and Atmospheric Administration (NOAA) (Pinzon and Tucker, 2014). The elevation dataset was obtained from the ETOPO Global Relief Model developed by the National Centers for Environmental Information (https://www.ncei.noaa.gov/products/etopo-global-relief-model). The aridity index dataset was derived from the Global Aridity Index and Potential Evapotranspiration Database—Version 3 (Zomer et al., 2022)." **(lines 112-136 of the revised manuscript)**

**Point #17**

*COMMENT: L140–143, some references to support these statements would be appreciated since the differences between a standardized approach and other methods for drought identification are well discussed in many papers now.*

*RESPONSE:* We sincerely appreciate the reviewer's helpful comment. Accordingly, in the revised manuscript, we have added references to support the statement regarding the advantages of SDI. In detail, the revised sentence is provided as follows:

"Compared with other drought indices, the SDI is not only simple and efficient to calculate, but also applicable to diverse climates due to its consistent calculation procedure (Zarch et al., 2015; Adnan et al., 2018)." **(lines 151-152 of the revised manuscript)**

**The added references:**

Adnan, S., Ullah, K., Shuanglin, L., Gao, S., Khan, A. H., and Mahmood, R.: Comparison of various drought indices to monitor drought status in Pakistan, Clim. Dynam., 51, 1885–1899, doi:10.1007/s00382-017-3987-0, 2018.
Zarch, M. A. A., Sivakumar, B., and Sharma, A.: Droughts in a warming climate: A global assessment of Standardized precipitation index (SPI) and Reconnaissance drought index (RDI), J. Hydrol., 526, 183–195, doi:10.1016/j.jhydrol.2014.12.065, 2015.

**Point #18**

*COMMENT: L154, which is the maximum accumulation period n that you tested? I assume 24 months from Fig. 1, but please specify.*

*RESPONSE:* We sincerely appreciate the reviewer's helpful comment. The 24 months is the maximum accumulation period for calculating the response time of drought propagation. In the updating manuscript, we have revised the sentence to make it clearer. The revised sentence is:

> "The correlation coefficient is calculated using Pearson's correlation coefficient formulated as follows (Pearson, 1896):
>
> $$r_P = \frac{\sum_{i=1}^{n}(x_i - \bar{x})(y_i - \bar{y})}{\sqrt{\sum_{i=1}^{n}(x_i - \bar{x})^2}\sqrt{\sum_{i=1}^{n}(y_i - \bar{y})^2}} \qquad (1)$$
>
> where $r_P$ represents the Pearson's correlation coefficient between SPI-n (n is the accumulation period, n = 1, 2, …, 24) and SSI-1; $\bar{x}$ and $\bar{y}$ represent the average value of SPI and SSI, respectively; $x_i$ and $y_i$ represents the SPI and SSI values in the time $i$, respectively." **(lines 162-167 of the revised manuscript)**

**Point #19**

*COMMENT: L163–166, clarification on why you applied these rules for drought selection would be appreciated.*

*RESPONSE:* We sincerely appreciate the reviewer's helpful comment. In this study, a multi-threshold run theory was employed to identify the drought events. This approach has advantages in avoiding the unreasonable splitting of persistent droughts and filtering out minor drought episodes, thus providing more accurate identification of drought events. Accordingly, we have revised the sentences to clarify why we use this method to identify drought events. The revised sentence is provided as follows:

> "Run theory is a commonly used method for analyzing drought characteristics, which defines the initiation and termination of a drought event based on the drought index. In this study, the drought events were identified using a multi-threshold run theory, which has advantages in avoiding the unreasonable splitting of persistent droughts and filtering out minor drought episodes, thus providing more accurate identification of drought events (Fleig et al., 2006; Ma et al., 2021). Potential drought events were initially identified using an intermediate threshold ($X_0 = 0$). Subsequently, the adjacent drought events with an interval of one month and whose drought index values were below a high threshold ($X_1 = 1$) within that month were combined. Finally, the potential drought events with one month length and whose drought index value is greater than a low threshold ($X_2 = -1$) were ruled out." **(lines 172-179 of the revised manuscript)**

**Point #20**

*COMMENT: (a) L195–196, I assume the factors that you use as model predictors regarding precipitation, temperature, potential evapotranspiration, runoff, soil moisture, and vegetation conditions are long-term averages, but I encourage you to specify this point in the text. If so, which period did you consider for averaging? (b) Also, why did you choose these specific factors? Other factors regarding e.g. soil or geology would also be important in my view. (c) In general, additional details on the models would be beneficial (e.g., training and validation periods, achieved model performances, etc).*

*RESPONSE:* We sincerely appreciate the reviewer's insightful comment. **(a)** In this study, the model predictors, including precipitation, temperature, potential evapotranspiration, runoff, soil moisture, and vegetation conditions, are long-term averages over the period 1958–2024. In the revised manuscript, we have rewritten the descriptions of the model predictors and target variables to improve clarity. **(b)** We agree with the reviewer's comment that there are a large number of factors that influence drought propagation, such as soil properties and geology factors. In our analysis, the selection of these factors as model predictors is due to the reason that (1) a large number of previous studies have demonstrated the importance of climatic factors in drought propagation (Apurv et al., 2017; Sattar et al., 2019; Apurv and Cai, 2020); (2) our research focused on the process of drought propagation at a $1°\times1°$ grid scale; however, soil properties and other geological factors are not easily aggregated at such a relatively coarse spatial resolution. **(c)** We thank the reviewer for this suggestion, which helps improve the reproducibility and transparency of our work. Accordingly, we have substantially expanded the Methods and Results section to emphasize the details of model development and evaluation. The revised sentences are provided as follows:

> "According to previous studies, climatic conditions are among the most important factors influencing drought propagation characteristics (Aryal et al., 2024). To explore the relative importance of long-term climatic conditions for drought propagation, the average values (1958–2024) of eight climatic and physiographic variables, including precipitation, temperature, potential evapotranspiration, runoff, soil moisture, aridity index, elevation, and vegetation condition, were selected as model predictors. The corresponding drought propagation characteristics (i.e., response time, propagation rate, and lag time) were selected as target variables. The Extreme Gradient Boosting (XGBoost) model was employed to model the relationships between climatic predictors and drought propagation target variables. The XGBoost model is an efficient and robust gradient-boosted decision tree algorithm that is widely applied in classification and regression tasks within the field of water resources engineering (Chen and Guestrin, 2016; Niazkar et al., 2024). To account for spatial autocorrelation, spatial block cross-validation was employed on the training set to prevent overfitting. The global grid was partitioned into 43 spatially contiguous blocks according to the IPCC AR6 reference land regions (Iturbide et al., 2020). In each fold, ten blocks were held out for validation, and the XGBoost model was trained on the remaining blocks. Model performance was evaluated using the coefficient of determination ($R^2$) and root mean square error (RMSE), averaged across all held-out blocks." **(lines 225-238 of the revised manuscript)**

**Point #21**

*COMMENT: Fig.1, caption, please expand the abbreviations (in other captions as well, in the*

*supplementary figures too). Also, some details are missing in this specific caption (e.g., inner plots - where axes are not labelled – and p value for statistical significance). I recommend you having another check that all information needed to fully understand the figures are reported in the captions, including those in the supplementary.*

***RESPONSE:*** We sincerely appreciate the reviewer's helpful comment. Accordingly, we have supplemented all the information required in the figures throughout the manuscript, including in the supplementary materials. The revised Figure 1 is provided as follows:

[Figure]

Figure 1. Spatial patterns of average response time from meteorological to hydrological droughts (RT$_{MH}$), from meteorological to agricultural droughts (RT$_{MA}$), and from hydrological to agricultural droughts (RT$_{HA}$), and the corresponding Pearson correlation coefficients derived from the ensemble of ERA5, GLDAS, and TerraClimate datasets. The blank grids indicate that the correlation between different drought indices is not statistically significant (p-value < 0.05). The inner plots show the histograms of response time and maximum correlation across global land areas.

**Point #22**

***COMMENT:*** *L212–213, it is not entirely clear to me what you mean by 'maximum correlation coefficients'. I suggest rephrasing. I would say that the robustness of your assessments comes from the relatively high correlation coefficients in Fig. 1 and their statistical significance.*

***RESPONSE:*** We sincerely appreciate the reviewer's insightful comment. In this study, the maximum correlation coefficients represent the highest value in the correlation analysis when the response time is identified. To avoid ambiguous expressions, we have rephrased the relevant

sentences to make it clearer. In detail, the revised sentences are provided as follows:

"The robustness of the response time evaluation can be attributed to the relatively high correlation coefficients presented in Fig. 1. The IQRs of corresponding correlation coefficients of SPI–SRI, SPI–SSI, and SRI–SSI are [0.43, 0.80], [0.51, 0.68], and [0.52, 0.70], respectively (Figs. 1B, 1D, and 1F). The correlation coefficients of response times are relatively high in the mid- to low-latitude regions (30 °S–30 °N), suggesting strong robustness of the response time measurements in these regions. The response times among meteorological, hydrological, and agricultural droughts also exhibit obvious seasonal variations (Figs. S1 and S2). Shorter response times and higher correlation coefficients were observed during the summer season (June–August in the Northern Hemisphere, and December–February in the Southern Hemisphere)." **(lines 252-260 of the revised manuscript)**

**Point #23**

**COMMENT:** *Fig. 2, wouldn't considering relative months from the start of the local water year rather than calendar months easier here? This would allow not to mix different processes occurring in the same months in the northern and southern hemispheres. Also, units for correlation coefficients are missing in the axes labels.*

**RESPONSE:** We sincerely appreciate the reviewer's insightful comment. We agree that the previous Fig. 2 is confusing because it conflates different processes occurring in the same month due to seasonal differences between the Northern and Southern Hemispheres. In the revised manuscript, the previous Fig. 2 has been moved to the supplementary material, and we have separated the results for the Northern and Southern Hemispheres into two different figures. In detail, the revised parts are provided as follows:

[Figure]

Figure S1. Box plots of $RT_{MH}$, $RT_{MA}$ and $RT_{HA}$ for each calendar month in the Southern Hemisphere, along with the corresponding Pearson correlation coefficients.

[Figure]

Figure S2. Box plots of $RT_{MH}$, $RT_{MA}$ and $RT_{HA}$ for each calendar month in the Northern Hemisphere, along with the corresponding Pearson correlation coefficients.

"The response times among meteorological, hydrological, and agricultural droughts also exhibit obvious seasonal variations (Figs. S1 and S2). Shorter response times and higher

correlation coefficients were observed during the summer season (June–August in the Northern Hemisphere, and December–February in the Southern Hemisphere)." **(lines 257-260 of the revised manuscript)**

**Point #24**

*COMMENT: Fig. 3, caption, which correlation is not statistically significant in blank grid cells? If I understand this analysis correctly (see comment #3), you computed multiple correlations here. Please specify.*

*RESPONSE:* We sincerely appreciate the reviewer's helpful comment. In the Fig. 3, the blank grid cell indicate that, within at least one time-window in the time series of response time obtained from the moving window, the correlation coefficient is not statistically significant. Accordingly, in the revised manuscript, we have added the relevant description to make it clearer. The revised figure is provided as follows:

[Figure]

Figure 3. Spatial patterns of time series trends in $RT_{MH}$, $RT_{MA}$ and $RT_{HA}$ across global land areas. The blank grids signify that, within at least one time-window in the time series of response time obtained from the moving window, the correlation coefficient is not statistically significant. The black dots indicate the statistical significance of the time series trend, where the p-value of the TFPW-MK test is less than 0.05. A significant increase (decrease) indicates that the Sen's slope is greater (less) than 0 and that the p-value of the TFPW-MK test is less than 0.05. A nonsignificant increase (decrease) indicates that the Sen's slope is greater (less) than 0 and that the p-value of the TFPW-MK test is greater than 0.05. A monotonic trend indicates that Sen's slope is equal

to 0.

**Point #25**

*COMMENT: (a) Fig. 4, which dataset does this figure refer to? The same comment applies also to other figures. I assume all the figures in the main text refer to the ensemble mean, but I would recommend specifying this somewhere. (b) With respect to this specific figure, I would also suggest correcting the label in the colour bars in panels a, c, and e to 'Propagation rate' rather than 'Response rate' for consistency with the rest of the manuscript and specifying in the caption the different y-axis in panel b as compared to panels d and f. (c) An additional comment on the analysis behind this figure: could the very low propagation rates in panels c and e be due to the time scale that you use? From my understanding, you are not considering any time lag between drought types, even though you show that some drought types can occur well after others in e.g., Fig. 1.*

*RESPONSE:* We sincerely appreciate the reviewer's insightful comment. **(a)** Fig. 4 presents the propagation rate and lag time derived from the ensemble mean of the ERA5, GLDAS, and TerraClimate datasets. In the revised manuscript, we have added an explanation of the datasets used in Figure 4. **(b)** We have corrected the label in the color bars in panels a, c, and e from "response rate" to "propagation rate". In addition, we have added an explanation of different y-axis in panel b as compared to panels d and f. In detail, the specific modifications are as follows:

[Figure]

Figure 4. Spatial patterns of propagation rate ($PR_{MH}$, $PR_{MA}$ and $PR_{HA}$) and lag time ($LT_{MH}$, $LT_{MA}$ and $LT_{HA}$) derived from the ensemble of ERA5, GLDAS, and TerraClimate datasets across global land areas. The inner plots show the histograms of propagation rate

and lag time across global land areas. The value of $LT_{MH}$ is lower those that of $LT_{MA}$ and $LT_{HA}$, so it is assigned a different color bar.

(c) In this study, the propagation rate and lag time are derived from discrete events identified through run theory analysis. In this process, meteorological, hydrological, and agricultural droughts were determined using the SPI, SRI, and SSI at a 1-month accumulation period. We agree with the reviewer's comment that, when considering the time lag between different drought types, the propagation rate among droughts may increase. However, this approach may obscure the actual correlations between different types of droughts. In the revised manuscript, we have rewritten the sentence in the section "2.4 Lag time analysis based on run theory" to clarify the accumulation period of drought indices in our study. In detail, the revised sentence is provided as follows:

> "To elucidate the propagation of drought across different types, the SPI, SRI, and SSI at a 1-month accumulation period were used to represent meteorological, hydrological, and agricultural drought, respectively. Consistent with the analysis of drought response time, we analyzed the propagation rate and lag time between meteorological and hydrological droughts ($PR_{MH}$ and $LT_{MH}$), between meteorological and agricultural droughts ($PR_{MA}$ and $LT_{MA}$), and between hydrologcial and agricultural droughts ($PR_{HA}$ and $LT_{HA}$)." **(lines 190-194 of the revised manuscript)**

**Point #26**

**COMMENT:** *Fig. 5, why are the blank areas here the same across the three maps? This is not the case in Fig. 3, which makes sense to me. Also, please add more information to this caption, in a similar way to what done for Fig. 3.*

**RESPONSE:** We sincerely appreciate the reviewer's insightful comment. Fig. 5 (Fig.7 in the updating manuscript) presents the time series trend of $PR_{MH}$, $PR_{MA}$ and $PR_{HA}$, which is calculated using the run theory. Unlike the lag time derived from correlation analysis that considered the statistical significance in Fig. 3, the propagation rate can be calculated from the grid with continuous meteorological and hydrological data. Therefore, the blank grids indicate that the data is missing in one of the ERA5, GLDAS, and TerraClimate datasets, which is same across the three maps. In the revised manuscript, we have added relevant explanations to clarify this content. In detail, the revised sentences are provided as follows:

> "The propagation rate and lag time derived from the runoff theory can be calculated from the grid using continuous meteorological and hydrological data; therefore, blank grids indicate missing data in at least one of the ERA5, GLDAS, or TerraClimate datasets." **(lines 367-369 of the revised manuscript)**

**Point #27**

**COMMENT:** *Fig. 6 shows decreasing trends in the lag time between meteorological and hydrological droughts in large portions of Europe and northern Asia that are not reflected in the propagation time though (Fig. 3). Do you have an explanation for such discrepancies? Also, I*

*see that attributing these trends to their causes might be outside of the scope of this current paper, but I think that some discussion on potential causes of these trends would still be valuable.*

**RESPONSE:** We sincerely appreciate the reviewer's insightful comment. The apparent discrepancy between the trends in response time (Fig. 3) and those in lag time (Fig. 7 in the revised manuscript) is indeed a critical observation, and we agree that discussing its potential causes significantly enhances the scientific value of the manuscript. Accordingly, we have added a Discussion section (Section 4.1: Physical Interpretation of Drought Propagation Characteristics) to analyze the reasons for the differences in lag time and response time. The revised paragraphs are provided as follows:

"4.1. Physical interpretation of drought propagation characteristics
In this study, two distinct methodological frameworks were employed to quantify drought propagation: (1) the response time derived from time-lag correlation analysis, and (2) the lag time based on event identification using the run theory. Response time is determined by identifying the accumulation period of a drought index (e.g., SPI) that maximizes its correlation with a target drought index (e.g., SSI at a 1-month accumulation timescale) (López-Moreno et al., 2013; Zhang et al., 2022). This approach reflects the overall synchronicity and statistical memory characteristics of various drought conditions. Thus, the response time values are strongly influenced by long-term variations in regional climatic and hydrological conditions, such as the seasonal cycle, multi-year climate oscillations, and water storage capacity. The response time refers to the system's long-term state that retains a memory of past drought conditions. The evaluation of response time is beneficial for seasonal drought predictability and long-term drought preparedness. The response time also functions as an indicator of the feasibility of using one type of drought index as a proxy for another. For example, due to the lack of comprehensive observational data, the SPI with varying accumulation periods can reflect hydrological, agricultural and groundwater drought conditions (Kumar et al., 2016).

In comparison, lag time is derived from discrete drought events identified using the multi-threshold run theory, which measures the time difference between the onset of one drought event and the onset of another drought event. By focusing on event-based dynamics, the lag time reflects the instantaneous triggering mechanism by which drought signals propagate from the atmosphere to the land surface. Numerous previous studies have analyzed the threshold of extreme stress that triggers drought propagation, using methods such as copula functions, hydrological models, and machine learning (Geng et al., 2024; Yang et al., 2025). The lag time captures the non-linear response mechanism between different drought conditions at a short time scale, which is crucial for real-time early warning and impact assessment.

Our results provide a globally consistent comparison of the response time and lag time for meteorological, hydrological, and agricultural drought propagation. The response time of drought propagation (average $RT_{MH}$, $RT_{MA}$, and $RT_{HA}$ of 5.0 [2.7, 6.7] months, 8.7 [5.0, 11.3] months, and 5.8 [2.3, 7.3] months) is generally longer than the lag time (average $LT_{MH}$, $LT_{MA}$, and $LT_{HA}$ of 1.23 [0.68, 1.68] months, 2.60 [1.71, 2.92] months, and 2.49 [1.68, 2.51] months). This numerical gap arises from differences in the methodology, but both approaches indicate a consistent propagation pathway for meteorological, hydrological, and agricultural droughts, with similar spatial patterns. In addition, the machine learning-based attribution method also identifies similar impact

factors, which indicates the consistency of drought propagation mechanisms revealed by response time and lag time. This aligns with the conceptual framework of drought propagation, where precipitation deficits (meteorological drought) first influence runoff generation over the land surface (hydrological drought), and subsequently affect soil moisture in the root zone (agricultural drought)." **(lines 437-468 of the revised manuscript)**

**Point #28**

*COMMENT: L327–328, 'the SHAP value indicates that high temperatures have shortened the response time of meteorological drought to hydrological drought', I stumbled a bit here. I suggest rephrasing, for instance by using present tenses, not to evoke changes over time, which is not what you are looking at with your SHAP-analysis. This comment applies also to L409–411.*

*RESPONSE:* We sincerely appreciate the reviewer's insightful comment. We agree with the reviewer's comment that the impacts of climatic and geographical factors are not reflected in the temporal changes observed in our analysis. In the revised manuscript, we have rephrased the sentences to avoid ambiguous descriptions. Specifically, the revised sentences are provided as follows:

"The meteorological-to-hydrological drought propagation characteristics are primarily influenced by regional temperature and PET, with the non-monotonic behaviour predominantly observed in the 30th to 70th percentiles of temperature and PET. In this percentile range, both $PT_{MH}$ and $LT_{MH}$ decrease as temperature and PET increase, while $PR_{MH}$ increases as temperature and PET increase." **(lines 423-426 of the revised manuscript)**

**Point #29**

*COMMENT: Fig. 9 and corresponding text, I would suggest turning 'key feature factors' to 'dominant factors' to enhance the clarity of which factors you are looking at here. Also, for Fig. 9, do you have an explanation for the non-monotonic behaviour of meteorological-to-hydrological drought propagation characteristics across different quantiles of the considered features?*

*RESPONSE:* We sincerely appreciate the reviewer's insightful comment. We agree that "dominant factors" is a more accurate description, expressing the main factors that influence drought propagation. For the meteorological-to-hydrological drought propagation characteristics, the non-monotonic behaviour predominantly observed in the 30th to 70th percentiles of temperature and PET. This is mainly due to the influence of temperature on the snow-related processes of the water cycle. Accordingly, we have added an explanation of the non-monotonic behavior of meteorological-to-hydrological drought propagation characteristics in the Results Analysis and Discussion section. In the revised manuscript, the updated sentences are as follows:

"The meteorological-to-hydrological drought propagation characteristics are primarily influenced by regional temperature and PET, with the non-monotonic behaviour

predominantly observed in the 30th to 70th percentiles of temperature and PET. In this percentile range, both $RT_{MH}$ and $LT_{MH}$ decrease as temperature and PET increase, while $PR_{MH}$ increases as temperature and PET increase." **(lines 423-426 of the revised manuscript)**

"Specifically, the non-monotonic behaviour of meteorological-to-hydrological drought propagation characteristics mainly occurred in the range of 20th to the 70th percentiles for temperatures and PET. In the subtropical regions, shorter $RT_{MH}$ and $LT_{MH}$ with low $PR_{MH}$ trend to occur in regions characterized by higher average temperature and PET (Fig. 11). This is primarily attributed to the influence of temperature on the snow-related processes of the water cycle, resulting in a delayed response of runoff to changes in precipitation. During cold seasons, precipitation is stored in the form of snow and ice in glaciers, which subsequently melt and contribute to runoff during the following warm season." **(lines 478-483 of the revised manuscript)**

**Point #30**

*COMMENT: L429–435, this part sounds to me more like a discussion of the implications rather than of uncertainties. I suggest moving to a new subsection or rename the current one.*

*RESPONSE:* We sincerely appreciate the reviewer's insightful comment. In the revised manuscript, we have renamed the subsection "4.4. Uncertainties and Implications for Drought Propagation Evaluation" to better highlight the implications of drought propagation. In detail, the revised paragraphs are provided as follows:

"4.4. Uncertainties and implications in drought propagation evaluation
Drought propagation evaluation relies heavily on drought indices for monitoring and characterizing various drought types. Considering the data availability and the continuity in both temporal and spatial dimensions at the global scale, we employed the SPI, SRI, and SSI to represent meteorological, hydrological, and agricultural droughts. Our results demonstrated the propagation pathway of meteorological-hydrological-agricultural droughts, which is consistent with previous studies that employed similar indices (Han et al., 2023; Mei et al., 2025). As a multifaceted phenomenon, hydrological drought is a broad term that is related not only to runoff but also to streamflow and the levels of groundwater, lakes, and reservoirs (Van Loon, 2015). Using the drought indices derived from streamflow, the propagation from agricultural to hydrological droughts has also been identified in many studies, particularly at the watershed scale (Odongo et al., 2023; Teutschbein et al., 2025). Runoff is the volume of water that originates from precipitation and flows over the land surface; it is not directly equal to the streamflow in stream channels. A deficit in runoff can directly affect the availability of soil moisture due to reduced recharge to the root zone, representing the propagation from hydrological drought to agricultural drought. In comparison, soil moisture retains precipitation that falls on the land surface and then delays the propagation time form precipitation to streamflow (McColl et al., 2017).

Due to the inherent variability of drought-related variables, significant uncertainties exist within hydrometeorological datasets (Bador et al., 2020). Our findings depend on an ensemble of three datasets (i.e., ERA5, GLDAS, and TerraClimate), which helps avoid biased and incomplete evaluations of drought propagation that could result from relying

on a single dataset. We conducted a comparative analysis of drought propagation characteristics derived from multiple datasets, systematically evaluating their consistency and discrepancies (Figs. 11-13). The results underscore the impact of input data uncertainties on the assessment of drought propagation, with notable discrepancies predominantly observed in the hyper-arid, high-latitude, and high-evaluation regions. This is primarily attributed to the scarcity of in-situ stations capable of providing continuous spatial and temporal observations in these regions. The data assimilation systems and land surface models employed across different datasets to fill missing observations inevitably introduce uncertainties in both model parameters and structural configurations.

Generally, our study provides a comprehensive assessment of multiple drought propagation characteristics across global land areas, which has significant implications for the development and improvement of drought monitoring and early warning systems. In tropical and sub-tropical regions, real-time monitoring of meteorological drought can improve the forecasting of hydrological drought; whereas in humid regions, drought indices based on precipitation and runoff could provide more accurate predictions of agricultural drought risks. Future research could focus on improving the accuracy of predicting future drought changes by incorporating the mechanisms of drought propagation into predictive models. In addition, human activities—such as water abstraction, reservoir regulation, and land-use change—can profoundly modify natural drought propagation processes by altering catchment storage and flow pathways, thereby influencing drought propagation. Future research could also focus on quantitatively disentangling the effects of human activities on drought propagation." **(lines 513-549 of the revised manuscript)**

**Point #31**

*COMMENT: L438–440, I suggest some rephrasing here to improve the clarity of this sentence.*

*RESPONSE:* We sincerely appreciate the reviewer's helpful comment. Accordingly, we have revised the Conclusion section in the revised manuscript to clarify the main contribution of our study. Specifically, the revised sentences are provided as follows:

"In this study, we systematically assessed the propagation characteristics of multiple drought types from 1958 to 2024 across global land areas. Based on an ensemble of multiple datasets (i.e., ERA5, GLDAS, and TerraClimate), three standardized drought indices (SDIs) derived from precipitation, runoff, and soil moisture were employed to represent meteorological, hydrological, and agricultural drought conditions, respectively. The lag time derived from correlation analysis, as well as the response time and propagation rate based on run theory, were examined to characterize the propagation of meteorological, hydrological, and agricultural droughts." **(lines 551-556 of the revised manuscript)**

**Point #32**

*COMMENT: L45, I cannot find the reference (Zhu et al., 2021) in the reference list, please add*

*it. In addition, include also Xiong et al. (2025), already cited in the text, among the global-scale studies?*

**RESPONSE:** We sincerely appreciate the reviewer's helpful comment. In the revised manuscript, the previous sentence has been rewritten, and the reference to Zhu et al. (2021) has been removed. In addition, the study by Xiong et al. (2025) is also a global-scale study; however, the relevant sentence in the updated manuscript has been rewritten, and Xiong et al. (2025) is not cited here.

**Point #33**

**COMMENT:** *L47, there are two entries for both (Yang et al., 2024) and (Shi et al., 2022) in the reference list. Which one are you referring to here? Please specify here and elsewhere in the manuscript.*

**RESPONSE:** We sincerely appreciate the reviewer's carefulness. In the revised manuscript, we have distinguished references with identical entries. The references of Yang et al. (2024) have been removed in the revised manuscript. Specifically, the revised sentences and corresponding references are provided below:

"For example, Shi et al. (2022a) examined the response time from meteorological and hydrological droughts using the maximum correlation coefficient method, and analyzed the variations in response time across different climatic regions." **(lines 52-54 of the revised manuscript)**

"Over the past decades, a large number of attribution studies have been conducted to quantify the impacts of climatic and geographical factors on drought propagation, using methods such as statistical analysis (Gevaert et al., 2018), clustering analysis (Liu et al., 2023), causality analysis (Shi et al., 2022b), and machine learning models (Muthuvel and Qin, 2025)." **(lines 72-75 of the revised manuscript)**

**The revised references:**

Shi, H., Zhou, Z., Liu, L., and Liu, S.: A global perspective on propagation from meteorological drought to hydrological drought during 1902–2014, Atmos. Res., 280, 106441, doi:10.1016/j.atmosres.2022.106441, 2022a.
Shi, H., Zhao, Y., Liu, S., Cai, H., and Zhou, Z.: A new perspective on drought propagation: causality. Geophys. Res. Lett., 49(2), e2021GL096758, doi:10.1029/2021GL096758, 2022b.

**Point #34**

**COMMENT:** *L127, I assume you mean here 'elevation' rather than 'evaluation'. Please correct it, here and throughout the manuscript.*

**RESPONSE:** We sincerely appreciate the reviewer's carefulness. This was a typographical error.

Accordingly, we have systematically corrected all instances where "elevation" was incorrectly used in place of "evaluation" throughout the entire manuscript.

**Point #35**

*COMMENT: L165, 'with on month' -> 'with one month'? Else, the sentence sounds strange to me. Please check.*

*RESPONSE:* We sincerely appreciate the reviewer's carefulness. This was a typographical error, and "with on month" should be corrected to "with one month". We apologize for this mistake and have revised the sentence in the updating manuscript.

**Point #36**

*COMMENT: L190, 'formula' -> 'formulated'?*

*RESPONSE:* We sincerely appreciate the reviewer's carefulness. We sincerely appreciate the reviewer's careful comment. This was a grammatical mistake, and "formula" should be corrected to "formulated". We apologize for this mistake and have revised the sentence in the updating manuscript.

**Point #37**

*COMMENT: L193, 'influencing on the model predictions' -> 'influencing the model predictions'*

*RESPONSE:* We sincerely appreciate the reviewer's carefulness. We sincerely appreciate the reviewer's careful comment. This was a grammatical mistake, and "influencing on the model predictions" should be corrected to "influencing the model predictions". We apologize for this mistake and have revised the sentence in the updating manuscript.

**Point #38**

*COMMENT: L282, 'the highest PRMH and LTMH values' -> the highest PRMH and lowest LTMH values?*

*RESPONSE:* We sincerely appreciate the reviewer's carefulness. This was a typographical error, and "the highest $PR_{MH}$ and $LT_{MH}$ values" should be corrected to "the highest $PR_{MH}$ and lowest $LT_{MH}$ values". We apologize for this mistake and have revised the sentence in the updating manuscript.

**Point #39**

***COMMENT:*** *Fig. 8 and 9, please correct the labels in panels c, f, and i with the subscript 'HA' instead of 'MA'.*

***RESPONSE:*** We sincerely appreciate the reviewer's carefulness. This was a typographical error in the figures, and we have corrected this mistake in the revised manuscript. Specifically, the revised figures are provided as follows:

[Figure]

Figure 10. Ranking of factors influencing drought propagation characteristics based on the absolute SHAP value.

[Figure]

Figure 11. Box plots of drought propagation characteristics across global land areas classified by the percentiles of dominant factors. The dominant factor is temperature for $RT_{MH}$, $PR_{MH}$, PET for $LT_{MH}$, and precipitation for the other characteristics.

**Point #40**

*COMMENT: L303–304, I suggest rephrasing or removing, since this wording does not sound fitting to this subsection to me.*

*RESPONSE:* We sincerely appreciate the reviewer's helpful comment. In the revised manuscript, we have removed this sentence.

**Point #41**

*COMMENT: L344–345, the first 'hydrological' should probably be 'agricultural'.*

*RESPONSE:* We sincerely appreciate the reviewer's carefulness. This was a typographical error, and we have revised the sentence as follows:

> "In comparison, precipitation serves as the main influencing factor in the propagation from both meteorological and hydrological droughts to agricultural drought." **(lines 411-413 of the revised manuscript)**

**Point #42**

*COMMENT:* L413, 'reasons' -> 'seasons'?

*RESPONSE:* We sincerely appreciate the reviewer's carefulness. This was a typographical error, and we have corrected this mistake in the revised manuscript.

**Point #43**

*COMMENT:* L424, Figures 11-13, these figures are reported in the supplement. Please correct.

*RESPONSE:* We sincerely appreciate the reviewer's carefulness. This was a typographical error, and we have corrected this mistake in the revised manuscript.

**Point #44**

*COMMENT:* L442, 'finding' -> 'findings'?

*RESPONSE:* We sincerely appreciate the reviewer's carefulness. This was a grammatical errors, and we have corrected this mistake in the revised manuscript.

Generally, we are deeply grateful to the reviewer's insight and careful review. His/her comments have greatly helped improve the paper. We also expressed our gratitude in the "**Acknowledgments**" of the revised manuscript.

---

## Author Comment (AC2)

**RESPONSES TO REVIEWER TWO'S COMMENTS**

We are grateful to Reviewer #2 for his/her insightful review. The provided comments have contributed substantially to improving the paper. According to them, we have made significant efforts to revise the manuscript, with the details explained as follows:

**Point #1**

**COMMENT:** *Main Critique on Methodology and Physical Interpretation: The study employs both the Response Time (RT) based on time-lag correlation and the Lag Time (LT) based on run-theory event identification to analyze drought propagation from the dual dimensions of statistical association and event evolution. However, I contend that there is a fundamental difference in their underlying physical mechanisms. RT reflects the overall synchronicity or "statistical memory" between long-series precipitation, runoff, and soil moisture. Its values are typically larger (e.g., 5–8 months in this study), primarily capturing the integrated system response driven by seasonal cycles, multi-year climate oscillations (e.g., ENSO), and the long-term water storage capacity of basins. In contrast, LT, based on discrete event tracking, focuses on the physical evolution of specific drought pulses within the hydrological cycle, reflecting the instantaneous triggering mechanism of drought signals penetrating from the atmosphere to the land surface; thus, its values are usually much smaller (e.g., 1.2–2.6 months).*

*The authors must go beyond simply listing these inconsistent indices in tables and provide a rigorous physical explanation for this "numerical gap" in the Discussion section. Specifically, does the long-period RT represent the smoothing effect of basin storage or seasonal cycles on drought signals, while the short-lived LT captures the non-linear rapid response mechanism when the system exceeds a threshold under extreme stress? Without clarifying why statistical correlation and event evolution differ so significantly in magnitude from a physical perspective, readers will find it difficult to judge which indices are more valuable for early warning, and may even question the robustness of the results. Therefore, I expect the authors to add a dedicated section in the revised manuscript to deeply discuss the physical coupling behind these methodological differences and clearly indicate how to weigh the use of these distinct propagation indices under different management needs.*

**RESPONSE:** We sincerely thank the reviewer's insightful and constructive comments regarding the methodological differences between response time (RT) and lag time (LT) and their physical implications. We fully agree with the comments that a deeper physical interpretation of the RT and LT results is essential to clarify the mechanism of drought propagation and to provide more valuable insights for drought risk management. In the revised manuscript, we have incorporated a dedicated subsection within the Discussion section (4.1. Physical interpretation of drought propagation characteristics) to explicitly address this issue. In detail, the revised section is provided as follows:

"4.1. Physical interpretation of drought propagation characteristics

In this study, two distinct methodological frameworks were employed to quantify drought propagation: (1) the response time derived from time-lag correlation analysis, and (2) the lag time based on event identification using the run theory. Response time is determined by identifying the accumulation period of a drought index (e.g., SPI) that maximizes its correlation with a target drought index (e.g., SSI at a 1-month accumulation timescale) (López-Moreno et al., 2013; Zhang et al., 2022). This approach reflects the overall synchronicity and statistical memory characteristics of various drought conditions. Thus, the response time values are strongly influenced by long-term variations in regional climatic and hydrological conditions, such as the seasonal cycle, multi-year climate oscillations, and water storage capacity. The response time refers to the system's long-term state that retains a memory of past drought conditions. The evaluation of response time is beneficial for seasonal drought predictability and long-term drought preparedness. The response time also functions as an indicator of the feasibility of using one type of drought index as a proxy for another. For example, due to the lack of comprehensive observational data, the SPI with varying accumulation periods can reflect hydrological, agricultural and groundwater drought conditions (Kumar et al., 2016).

In comparison, lag time is derived from discrete drought events identified using the multi-threshold run theory, which measures the time difference between the onset of one drought event and the onset of another drought event. By focusing on event-based dynamics, the lag time reflects the instantaneous triggering mechanism by which drought signals propagate from the atmosphere to the land surface. Numerous previous studies have analyzed the threshold of extreme stress that triggers drought propagation, using methods such as copula functions, hydrological models, and machine learning (Geng et al., 2024; Yang et al., 2025). The lag time captures the non-linear response mechanism between different drought conditions at a short time scale, which is crucial for real-time early warning and impact assessment.

Our results provide a globally consistent comparison of the response time and lag time for meteorological, hydrological, and agricultural drought propagation. The response time of drought propagation (average $RT_{MH}$, $RT_{MA}$, and $RT_{HA}$ of 5.0 [2.7, 6.7] months, 8.7 [5.0, 11.3] months, and 5.8 [2.3, 7.3] months) is generally longer than the lag time (average $LT_{MH}$, $LT_{MA}$, and $LT_{HA}$ of 1.23 [0.68, 1.68] months, 2.60 [1.71, 2.92] months, and 2.49 [1.68, 2.51] months). This numerical gap arises from differences in the methodology, but both approaches indicate a consistent propagation pathway for meteorological, hydrological, and agricultural droughts, with similar spatial patterns. In addition, the machine learning-based attribution method also identifies similar impact factors, which indicates the consistency of drought propagation mechanisms revealed by response time and lag time. This aligns with the conceptual framework of drought propagation, where precipitation deficits (meteorological drought) first influence runoff generation over the land surface (hydrological drought), and subsequently affect soil moisture in the root zone (agricultural drought)." **(lines 435-466 of the revised manuscript)**

**Point #2**

**COMMENT:** *The paper analyzes three pathways: M→H, M→A, and H→A. To what extent is the propagation of H→A independent of M? That is, if meteorological drought (M) has already*

*directly driven agricultural drought (A), is the contribution of hydrological drought (H) to A merely a "shadow" of M?*

*RESPONSE:* We sincerely appreciate the reviewer's insightful comment. In this study, we analyzed three drought propagation pathways: from meteorological to hydrological drought, from meteorological to agricultural drought, and from hydrological to agricultural drought. Generally, drought propagation is regarded as a hierarchical top-down process. Our results demonstrate the pathway of meteorological-hydrological agricultural droughts. Meteorological drought, primarily caused by precipitation deficits, can cascade to other hydrological variables in the water cycle. As defined by the runoff variations, hydrological droughts are influenced by meteorological droughts and then propagate to agricultural droughts (deficits in soil moisture). We agree the reviewer's comment that the contribution of hydrological drought to agricultural drought is influenced by the meteorological drought. In fact, drought propagation is a complex process, as it is driven by the close interrelationships among various hydrological variables. The current analysis in our manuscript is hardly to distinguish the propagation of hydrological to agricultural droughts that is independent of the impact of meteorological drought. Accordingly, in the revised manuscript, we have added the discussion about the uncertainties in our analysis. Specifically, the revised paragraphs are provided as follows:

> "4.4. Uncertainties and implications in drought propagation evaluation
> Drought propagation evaluation relies heavily on drought indices for monitoring and characterizing various drought types. Considering the data availability and the continuity in both temporal and spatial dimensions at the global scale, we employed the SPI, SRI, and SSI to represent meteorological, hydrological, and agricultural droughts. Our results demonstrated the propagation pathway of meteorological-hydrological-agricultural droughts, which is consistent with previous studies that employed similar indices (Han et al., 2023; Mei et al., 2025). As a multifaceted phenomenon, hydrological drought is a broad term that is related not only to runoff but also to streamflow and the levels of groundwater, lakes, and reservoirs (Van Loon, 2015). Using the drought indices derived from streamflow, the propagation from agricultural to hydrological droughts has also been identified in many studies, particularly at the watershed scale (Odongo et al., 2023; Teutschbein et al., 2025). Runoff is the volume of water that originates from precipitation and flows over the land surface; it is not directly equal to the streamflow in stream channels. A deficit in runoff can directly affect the availability of soil moisture due to reduced recharge to the root zone, representing the propagation from hydrological drought to agricultural drought. In comparison, soil moisture retains precipitation that falls on the land surface and then delays the propagation time form precipitation to streamflow (McColl et al., 2017)." **(lines 513-526 of the revised manuscript)**

**Point #3**

*COMMENT: In desert regions with extremely low precipitation, the correlation between SPI and SRI is often meaningless. How were these extreme climatic zones handled in your global assessment, and are the conclusions applicable there?*

*RESPONSE:* We sincerely appreciate the reviewer's insightful comment. We agree that in hyper-arid regions—where precipitation is extremely low and highly erratic—the calculation of

the SPI and SRI becomes statistically unstable, and the correlation between SPI and SRI in such environments can indeed be uninterpretable. Accordingly, we have added a discussion of the uncertainties associated with drought propagation in hyper-arid regions. The revised paragraph is provided below:

> "Due to the inherent variability of drought-related variables, significant uncertainties exist within hydrometeorological datasets (Bador et al., 2020). Our findings depend on an ensemble of three datasets (i.e., ERA5, GLDAS, and TerraClimate), which helps avoid biased and incomplete evaluations of drought propagation that could result from relying on a single dataset. We conducted a comparative analysis of drought propagation characteristics derived from multiple datasets, systematically evaluating their consistency and discrepancies (Figs. S3-S6). The results underscore the impact of input data uncertainties on the assessment of drought propagation, with notable discrepancies predominantly observed in the hyper-arid, high-latitude, and high-evaluation regions. Specifically, in hyper-arid regions—where precipitation is extremely low and highly erratic—the calculation of the SPI and SRI becomes statistically unstable; consequently, the correlation between SPI and SRI in such environments can indeed be uninterpretable. This is primarily attributed to the scarcity of in-situ stations capable of providing continuous spatial and temporal observations in these regions. The data assimilation systems and land surface models employed across different datasets to fill missing observations inevitably introduce uncertainties in both model parameters and structural configurations." **(lines 528-539 of the revised manuscript)**

**Point #4**

*COMMENT: The authors selected eight factors for attribution. What was the rationale for selecting these specific factors? Why were underlying surface characteristics, such as land use types, not included? These physical surface features often have a more direct impact on drought propagation (especially PR and LT) than NDVI.*

*RESPONSE:* We sincerely appreciate the reviewer's insightful comment. We agree with the reviewer's comment that there are a large number of factors that influence drought propagation, such as soil properties and geology factors. In our analysis, the selection of these factors as model predictors is due to the reason that (1) a large number of previous studies have demonstrated the importance of climatic factors in drought propagation (Apurv et al., 2017; Sattar et al., 2019; Apurv and Cai, 2020); (2) our research focused on the process of drought propagation at a $1°\times1°$ grid scale; however, soil properties and other geological factors are not easily aggregated at such a relatively coarse spatial resolution. Accordingly, we have substantially expanded the Methods and Results section to emphasize the details of model development and evaluation. The revised sentences are provided as follows:

> "According to previous studies, climatic conditions are among the most important factors influencing drought propagation characteristics (Aryal et al., 2024). To explore the relative importance of long-term climatic conditions for drought propagation, the average values (1958–2024) of eight climatic and physiographic variables, including precipitation, temperature, potential evapotranspiration, runoff, soil moisture, aridity index, elevation, and vegetation condition, were selected as model predictors. The corresponding drought

propagation characteristics (i.e., response time, propagation rate, and lag time) were selected as target variables. The Extreme Gradient Boosting (XGBoost) model was employed to model the relationships between climatic predictors and drought propagation target variables. The XGBoost model is an efficient and robust gradient-boosted decision tree algorithm that is widely applied in classification and regression tasks within the field of water resources engineering (Chen and Guestrin, 2016; Niazkar et al., 2024). To account for spatial autocorrelation, spatial block cross-validation was employed on the training set to prevent overfitting. The global grid was partitioned into 43 spatially contiguous blocks according to the IPCC AR6 reference land regions (Iturbide et al., 2020). In each fold, ten blocks were held out for validation, and the XGBoost model was trained on the remaining blocks. Model performance was evaluated using the coefficient of determination ($R^2$) and root mean square error (RMSE), averaged across all held-out blocks." **(lines 225-238 of the revised manuscript)**

**Point #5**

**COMMENT:** *Global grid data exhibit strong spatial autocorrelation. If all grid points are fed directly into the XGBoost model, the model may suffer from overfitting or yield erroneous significance levels. Have the authors attempted to prove the robustness of the model?*

**RESPONSE:** We sincerely appreciate the reviewer's insightful comment. We agree that global gridded data exhibit strong spatial autocorrelation, which can lead to overfitting and thus reduce the generalizability of our findings. In our analysis, spatial block cross-validation was employed to account for spatial autocorrelation. The global grids were partitioned into 43 spatially contiguous blocks according to the IPCC AR6 reference land regions (Iturbide et al., 2020). In each fold, ten blocks were held out for validation, and the XGBoost model was trained on the remaining blocks. In the updating manuscript, we have added the sentence in the Methods section to make it clearer. The revised sentences are provided as follows:

"According to previous studies, climatic conditions are among the most important factors influencing drought propagation characteristics (Aryal et al., 2024). To explore the relative importance of long-term climatic conditions for drought propagation, the average values (1958–2024) of eight climatic and physiographic variables, including precipitation, temperature, potential evapotranspiration, runoff, soil moisture, aridity index, elevation, and vegetation condition, were selected as model predictors. The corresponding drought propagation characteristics (i.e., response time, propagation rate, and lag time) were selected as target variables. The Extreme Gradient Boosting (XGBoost) model was employed to model the relationships between climatic predictors and drought propagation target variables. The XGBoost model is an efficient and robust gradient-boosted decision tree algorithm that is widely applied in classification and regression tasks within the field of water resources engineering (Chen and Guestrin, 2016; Niazkar et al., 2024). To account for spatial autocorrelation, spatial block cross-validation was employed on the training set to prevent overfitting. The global grid was partitioned into 43 spatially contiguous blocks according to the IPCC AR6 reference land regions (Iturbide et al., 2020). In each fold, ten blocks were held out for validation, and the XGBoost model was trained on the remaining blocks. Model performance was evaluated using the coefficient of determination ($R^2$) and root mean square error (RMSE), averaged across all held-out blocks." **(lines 225-238 of the revised manuscript)**

"Using different drought propagation characteristics as the target variables, nine XGBoost models were established. The validation sets of these models yielded satisfactory evaluation results (Table S1), which can substantiate the attribution results." **(lines 385-387 of the revised manuscript)**

**The added table:**

Table S1. The performance metrics in validation sets for each XGBoost model

| Model name | $RT_{MH}$ | $RT_{MA}$ | $RT_{HA}$ | $PR_{MH}$ | $PR_{MA}$ | $PR_{HA}$ | $LT_{MH}$ | $LT_{MA}$ | $LT_{HA}$ |
|---|---|---|---|---|---|---|---|---|---|
| $R^2$ | 0.653 | 0.878 | 0.858 | 0.824 | 0.944 | 0.913 | 0.646 | 0.581 | 0.652 |
| RMSE | 1.736 | 1.779 | 1.667 | 4.613 | 3.252 | 4.416 | 1.177 | 4.221 | 1.411 |

**Point #6**

*COMMENT: In the Introduction, please emphasize that meteorological, hydrological, and agricultural systems are not isolated but are coupled through the hydrological cycle.*

*RESPONSE:* We sincerely appreciate the reviewer's valuable suggestion. We fully agree that meteorological, hydrological, and agricultural droughts are interconnected through the hydrological cycle—a connection that provides a stronger and more physically grounded foundation for our study. Accordingly, we have revised the Introduction in the updated version to incorporate this conceptual emphasis. The revised sentences are provided as follows:

"There exists a strong interrelationship among different types of droughts, owing to the close linkage of their driving factors within the hydrological cycle." **(lines 43-45 of the revised manuscript)**

**Point #7**

*COMMENT: Add a mention of the "Propagation Threshold" in the Introduction.*

*RESPONSE:* We sincerely appreciate the reviewer's helpful comment. Propagation threshold is an important concept for understanding drought propagation dynamics. Accordingly, we have incorporated this concept in the Introduction, and the revised sentences are provided as follows:

"Understanding drought propagation characteristics, such as propagation time, probability, and threshold, are essential for elucidating drought occurrence and evolution mechanisms, which help facilitate the effective drought monitoring and early warning systems." **(lines 46-49 of the revised manuscript)**

**Point #8**

***COMMENT:*** *In the Data section, the spatial resolution of different datasets should be clarified, and any resampling operations must be mentioned.*

***RESPONSE:*** We sincerely appreciate the reviewer's helpful comment. In this study, three different datasets were employed to assess the drought propagation characteristics, which have different spatial resolutions. To ensure spatial and temporal consistency, the period from 1958 to 2024 was selected as the reference period, and all datasets were uniformly interpolated onto a 1 °×1 °latitude–longitude grid using bilinear interpolation. In the updating version, we have rewritten the section "2.1 Datasets" to make it clearer, and the revised parts are provided as follows:

"2.1 Datasets
Monthly precipitation, runoff, and soil moisture were derived from the ERA5, the Global Land Data Assimilation System (GLDAS), and TerraClimate datasets to calculate the drought indices. ERA5 is the fifth-generation global atmospheric reanalysis product developed by the European Centre for Medium-Range Weather Forecasts. It integrates extensive records of both in-situ and satellite observations through an ensemble-based data assimilation system (Hersbach et al., 2020). Precipitation in ERA5 was generated by the atmospheric component of the Integrated Forecasting System, whereas runoff and soil moisture were simulated by a land surface model (Boussetta et al., 2021). The soil moisture in ERA5 was aggregated to 1 meter volumetric soil water using weighted data from three layers: 0–7 cm, 7–28 cm, and 28–100 cm. GLDAS is a multi-model ensemble comprising three land surface models—Noah, Catchment, and the Variable Infiltration Capacity—which integrate satellite and in-situ observations through advanced land surface modeling techniques. The soil moisture in GLDAS models has different soil layer structures, all of which were weighted to the root zone depth of 1 meter to be consistent with ERA5. TerraClimate integrates multiple datasets, including WorldClim, Climate Research Unit, and Japanese 55-year Reanalysis, to generate hydro-meteorological variables (Abatzoglou et al., 2018). The soil moisture in the TerraClimate refers to the plant extractable soil water based on the root zone storage capacity, as modeled by an empirical water balance model. To ensure spatial and temporal consistency, the period from 1958 to 2024 was selected as the reference period, and all datasets were uniformly interpolated onto a 1 °×1 °latitude–longitude grid using bilinear interpolation.

In addition, the temperature and potential evapotranspiration (PET) were also obtained from the ensemble of ERA5, GLDAS, and TerraClimate datasets. Potential evapotranspiration in these datasets was calculated using the Penman-Monteith method (Abatzoglou et al., 2018). The Normalized Difference Vegetation Index (NDVI) was obtained directly from the Advanced Very High Resolution Radiometer instruments operated by the National Oceanic and Atmospheric Administration (NOAA) (Pinzon and Tucker, 2014). The elevation dataset was obtained from the ETOPO Global Relief Model developed by the National Centers for Environmental Information (https://www.ncei.noaa.gov/products/etopo-global-relief-model). The aridity index dataset was derived from the Global Aridity Index and Potential Evapotranspiration Database—Version 3 (Zomer et al., 2022)." **(lines 112-136 of the revised manuscript)**

**Point #9**

*COMMENT: More details need to be added to Section 2.4.*

*RESPONSE:* We sincerely appreciate the reviewer's helpful comment. In the revised manuscript, we have added more details about the lag time and propagation rate derived from the multi-threshold run theory. Specifically, the revised paragraphs are provided as follows:

"2.4 Lag time analysis based on run theory
Run theory is a commonly used method for analyzing drought characteristics, which defines the initiation and termination of a drought event based on the drought index. In this study, the drought events were identified using a multi-threshold run theory, which has advantages in avoiding the unreasonable splitting of persistent droughts and filtering out minor drought episodes, thus providing more accurate identification of drought events (Fleig et al., 2006; Ma et al., 2021). Potential drought events were initially identified using an intermediate threshold ($X_0 = 0$). Subsequently, the adjacent drought events with an interval of one month and whose drought index values were below a high threshold ($X_1 = 1$) within that month were combined. Finally, the potential drought events with one month length and whose drought index value is greater than a low threshold ($X_2 = -1$) were ruled out.

After using run theory to identify the initiation and termination of drought events, the propagation rate and lag time between the two types of droughts can be evaluated. Taking meteorological and agricultural droughts as an example, the propagation from meteorological drought to agricultural drought is defined as the occurrence of an agricultural drought event during the period in which a meteorological drought occurs. Thus, the propagation rate ($PR_{MA}$) and lag time ($LT_{MA}$) can be mathematically expressed as follows (Sattar et al., 2019):

$$P_{MA} = \frac{n}{m} \times 100\% \qquad (2)$$

$$LT_{MA} = \frac{\sum_{i=1}^{n}(T_{M,i} - T_{A,i})}{n} \qquad (3)$$

where n is number of meteorological drought events that propagate to agricultural drought events; m is the total number of meteorological drought events during the study period; TM,i is the starting time of meteorological drought event i, and TA,i is the starting time of agricultural drought event i. To elucidate the propagation of drought across different types, the SPI, SRI, and SSI at a 1-month accumulation period were used to represent meteorological, hydrological, and agricultural drought, respectively. Consistent with the analysis of drought response time, we analyzed the propagation rate and lag time between meteorological and hydrological droughts ($PR_{MH}$ and $LT_{MH}$), between meteorological and agricultural droughts ($PR_{MA}$ and $LT_{MA}$), and between hydrologcial and agricultural droughts ($PR_{HA}$ and $LT_{HA}$)." **(lines 171-189 of the revised manuscript)**

**Point #10**

*COMMENT: It is suggested to add a brief explanation of "Non-significant areas" in Section 3.1.*

*RESPONSE:* We sincerely appreciate the reviewer's helpful comment. In the revised manuscript, we have added a brief explanation of non-significant trend of time series trend in the Fig. 3. In detail, the revised sentences is provided as follows:

[Figure]

Figure 3. Spatial patterns of time series trends in $RT_{MH}$, $RT_{MA}$ and $RT_{HA}$ across global land areas. The blank grids signify that, within at least one time-window in the time series of response time obtained from the moving window, the correlation coefficient is not statistically significant. The black dots indicate the statistical significance of the time series trend, where the p-value of the TFPW-MK test is less than 0.05. A significant increase (decrease) indicates that the Sen's slope is greater (less) than 0 and that the p-value of the TFPW-MK test is less than 0.05. A nonsignificant increase (decrease) indicates that the Sen's slope is greater (less) than 0 and that the p-value of the TFPW-MK test is greater than 0.05. A monotonic trend indicates that Sen's slope is equal to 0.

**Point #11**

*COMMENT: Figure 2 uses a unified global timeline. Since seasons are opposite in the Northern and Southern Hemispheres, the high response values in February–April might be entirely driven by the Northern Hemisphere. Should these be discussed separately?*

*RESPONSE:* We sincerely appreciate the reviewer's insightful comment. We agree that drought propagation exhibits distinct seasonal variations and differs between the Northern and Southern Hemispheres. Accordingly, we have split the original Fig. 2 into separate panels for the Northern and Southern Hemispheres to better illustrate these hemispheric differences in seasonal patterns.

In the revised manuscript, the original Fig. 2 has been moved to the Supplementary Materials, and the revised version is presented below:

[Figure]

Figure S1. Box plots of RT$_{MH}$, RT$_{MA}$ and RT$_{HA}$ for each calendar month in the Southern Hemisphere, along with the corresponding Pearson correlation coefficients.

[Figure]

Figure S2. Box plots of RT$_{MH}$, RT$_{MA}$ and RT$_{HA}$ for each calendar month in the Northern Hemisphere, along with the corresponding Pearson correlation coefficients.

"The response times among meteorological, hydrological, and agricultural droughts also exhibit obvious seasonal variations (Figs. S1 and S2). Shorter response times and higher correlation coefficients were observed during the summer season (June–August in the Northern Hemisphere, and December–February in the Southern Hemisphere)." **(lines 257-260 of the revised manuscript)**

**Point #12**

*COMMENT: Human activities can significantly alter the PR and LT of drought propagation. Have the authors considered quantifying human activities? Although this is challenging, I suggest a rough discussion on this topic.*

*RESPONSE:* We sincerely appreciate the reviewer's insightful comment. We agree that human activities—such as water abstraction, reservoir regulation, and land-use change—can profoundly modify natural drought propagation processes by altering catchment storage and flow pathways, thereby influencing both the propagation rate (PR) and lag time (LT). In our analysis, we focused on understanding drought propagation under the predominant influence of climate and natural conditions. Quantitatively disentangling the effect of human activities on drought propagation is indeed exceptionally challenging due to the scarcity of consistent, high-resolution datasets on human activities, particularly at the global scale. In response to the reviewer's comment, we have added a paragraph in the Discussion section addressing this topic, and the revised paragraphs are provided as follows:

"In addition, human activities—such as water abstraction, reservoir regulation, and land-use change—can profoundly modify natural drought propagation processes by altering catchment storage and flow pathways, thereby influencing drought propagation. Future research could also focus on quantitatively disentangling the effects of human activities on drought propagation." **(lines 546-549 of the revised manuscript)**

**Point #13**

*COMMENT: I strongly recommend placing the propagation maps generated by each individual dataset in the Supplementary Materials.*

*RESPONSE:* We sincerely appreciate the reviewer's helpful comment. Accordingly, we have added the propagation maps derived from different datasets in the Supplementary Materials. In detail, the revised parts are provided as follows:

[Figure]

Figure S3. Spatial patterns of average $RT_{MH}$, $RT_{MA}$, and $RT_{HA}$ across global land areas in the ERA5, GLDAS, and TerraClimate datasets.

[Figure]

Figure S4. Spatial patterns of maximum Pearson correlation coefficients for $RT_{MH}$, $RT_{MA}$, and $RT_{HA}$ across global land areas in the ERA5, GLDAS, and TerraClimate datasets.

[Figure]

Figure S5. Spatial patterns of average $PR_{MH}$, $PR_{MA}$, and $PR_{HA}$ across global land areas in the ERA5, GLDAS, and TerraClimate datasets.

[Figure]

Figure S6. Spatial patterns of average $LT_{MH}$, $LT_{MA}$, and $LT_{HA}$ across global land areas in the ERA5, GLDAS, and TerraClimate datasets.

Generally, we are deeply grateful to the reviewer's insight and careful review. His/her comments have greatly helped improve the paper. We also expressed our gratitude in the "**Acknowledgments**" of the revised manuscript.